# Spatiotemporal Dynamics of Direct Carbon Emission and Policy Implication of Energy Transition for China's Residential Consumption Sector by the Methods of Social Network Analysis and Geographically Weighted Regression

Yuling Sun [1,2], Junsong Jia [1,2,*], Min Ju [1,2] and Chundi Chen [3]

[1] Key Laboratory of Poyang Lake Wetland and Watershed Research, Ministry of Education, School of Geography and Environment, Jiangxi Normal University, Nanchang 330022, China; sunyuling@jxnu.edu.cn (Y.S.); jxsddljm@jxnu.edu.cn (M.J.)
[2] Graduate School, Jiangxi Normal University, Nanchang 330022, China
[3] College of Architecture and Urban Planning, Tongji University, Shanghai 200092, China; chundichen@tongji.edu.cn
[*] Correspondence: jiaaniu@jxnu.edu.cn

**Abstract:** As China's second largest energy-use sector, residential consumption has a great potential for carbon dioxide ($CO_2$) reduction and energy saving or transition. Thus, here, using the methods of social network analysis (SNA) and geographically weighted regression (GWR), we investigated the spatiotemporal evolution characteristics of China's residential $CO_2$ emissions (RCEs) from direct energy use and proposed some policy suggestions for regional energy transition. (1) From 2000 to 2019, the total direct RCEs rose from 396.32 Mt to 1411.69 Mt; the consumption of electricity and coal were the primary sources. Controlling coal consumption and increasing the proportion of electricity generated from renewable energy should be the effective way of energy transition. (2) The spatial associations of direct RCEs show an obvious spatial network structure and the number of associations is increasing. Provinces with a higher level of economic development (Beijing, Shanghai, and Jiangsu) were at the center of the network and classified as the net beneficiary cluster in 2019. These provinces should be the priority areas of energy transition. (3) The net spillover cluster (Yunnan, Shanxi, Xinjiang, Gansu, Qinghai, Guizhou) is an important area to develop clean energy. People in this cluster should be encouraged to use more renewable energy. (4) GDP and per capita energy consumption had a significant positive influence on the growth of direct RCEs. Therefore, the national economy should grow healthily and sustainably to provide a favorable economic environment for energy transition. Meanwhile, residential consumption patterns should be greener to promote the use of clean energy.

**Keywords:** energy transition; residential $CO_2$ emissions (RCEs); social network analysis; geographically weighted regression; spatial association network

## 1. Introduction

With the growth of the world's population and economy, global energy consumption has increased dramatically since the industrial revolution [1]. Much energy consumption can inevitably emit a large number of greenhouse gases (GHGs), especially carbon dioxide ($CO_2$). Thus, global climate change (warming) has almost become an undeniable fact [2,3]. At present, the adverse effects of climate change are becoming increasingly apparent [4]. How to deal with climate change is the issue of our time. China is the biggest energy user and the largest $CO_2$ emitter, its $CO_2$ emissions accounted for 27.8% of total $CO_2$ worldwide in 2018 [5]. The total $CO_2$ emissions of China will keep growing under the influence of continued economic growth [6,7]. Therefore, a series of targets and policies were formulated by China to mitigate emissions under huge pressure and in the context of

low-carbon socio-ecological transition. Currently, China aims to have peak $CO_2$ emissions by 2030 and work towards carbon neutrality by 2060 [8]. To ensure that the goal of carbon peaking by 2030 will be met on schedule, a plan for carbon peaking by 2030 was issued. It proposed that the proportion of non-fossil energy consumption should reach about 20% by 2025 and about 25% by 2030. The plan also focused on the implementation of emissions mitigation and energy transition, even elevating it to a national action [9].

Energy consumption is closely related to $CO_2$ emissions [10,11]. From 2000 to 2019, the difference in total energy consumption between the residential sector and industrial sector is huge. The industry sector has been the largest energy consumption sector; the residential sector has been the second largest energy consumption sector (Figure 1). The residential sector is the non-negligible source of $CO_2$ emissions [12]. Some studies found that $CO_2$ emissions from residential energy consumption even surpassed the $CO_2$ emissions of industry in some developed countries [13,14]. Additionally, the rising disposable income of China's residents [15] and acceleration of urbanization [16,17] have made residents' lifestyles higher in carbonization [18–20], which inevitably increased the amount of $CO_2$ emissions from residential energy consumption (RCEs). Consequently, RCEs have attracted scholars' extensive attention. Wei et al. quantified the direct and indirect impact of lifestyle on China's energy use and the related $CO_2$ emissions [21]. Li et al. investigated the impact of social awareness on China's residential carbon emissions [22]. Ma et al. calculated carbon emissions from residential energy consumption in China and the USA [23]. However, consideration of spatial effects for RCEs in these studies was insufficient. With the depth of research on RCEs and the development of spatial measurement techniques, the spatial effect of RCEs is gaining attention. Zhou et al. used spatial autocorrelation analysis to analyze changes in the spatial pattern of residential carbon emissions [24]. Long et al. also used spatial autocorrelation analysis and investigated the spatiotemporal variation of $CO_2$ emissions generated by household private cars [25]. Using spatial autocorrelation analysis, Rong et al. investigated the spatial autocorrelation of RCEs in Kaifeng [26]. However, there are limitations in the previous studies on spatial effects in that these studies just considered an attribute rather than a relationship and believed only geographically adjacent areas to be correlated; the spatial association is always regarded as geographic adjacency. Therefore, social network analysis (SNA) was proposed to quantify network actors and their connections using relational data instead of traditional attribute data. Moreover, it also describes the characteristics of network associations and interactions between actors. SNA has been widely used to study public opinion [27], population migration [28], and tourism activities [29]. It has also recently begun to receive attention in the energy consumption field and $CO_2$ emissions. Bu et al. innovatively used SNA to reveal the network characteristics and spatial patterns of interprovincial natural gas consumption in China [30]. He et al. and Bai et al. used SNA, respectively, to explore the spatial association network characteristics of China's electricity sector [31] and transportation sector [32]. Shen et al. combined disparity analysis and SNA to investigate the synergistic emissions reduction effect of urban agglomerations [33]. However, SNA has rarely been applied in the study of RCEs.

The RCEs can be divided into direct and indirect emissions [21]. Direct RCEs are the $CO_2$ emissions that come directly from residents consuming fossil energy and secondary energy in activities such as lighting, cooking, and travel by personal transport [34,35]. Indirect RCEs are induced by the energy use of non-energy products consumed by residents in clothing, food, housing, and transportation for all life-cycle links [36]. Numerous previous studies have analyzed the factors affecting residential $CO_2$ emissions by various methods. Li et al. used the logarithmic mean Divisia index (LMDI) to identify the driving factors of RCEs [37]. Wang et al. used the stochastic impacts by regression on population, affluence, and technology (STIRPAT) model to analyze the factors influencing total carbon emissions of households [38]. Zang et al. examined the driving factors behind household direct $CO_2$ emissions by the LMDI method [39]. Fan et al. used the adaptive weighting Divisia (AWD) to identify the quantitative effects of the driving components [40]. Using

the LMDI model, Zhou et al. explored the driving factors behind household indirect $CO_2$ emissions [41]. Yuan et al. proposed a new structural decomposition analysis (SDA) model to investigate regional variations in factors' impacts on indirect RCEs in China [42]. These studies on drivers mainly used the econometric method to decompose the driving factors simply and the investigation of differences in the effects of drivers by spatial location was inadequate. $CO_2$ emissions, as an atmospheric resource, are changeable in different geographical locations [43] and the effects of $CO_2$ emissions correspondingly vary at different locations. For this reason, it is essential to study the variation of $CO_2$ emissions and their driving factors in different geographical locations (the spatial heterogeneity of $CO_2$ emissions and their driving factors). The geographically weighted regression (GWR) model is one useful method to deal with spatial heterogeneity. The estimation results of GWR not only consider the drivers' spatial location but also incorporate the data's spatial characteristics. Sheng et al. found that GWR is more appropriate to estimate parameters than ordinary least square (OLS) in a study of $CO_2$ emissions [44]. Wang et al. came to the same conclusion [45]. Moreover, Wang et al. investigated the effect of urbanization on $CO_2$ emissions in China's six sectors. They found that the influence of urbanization has significant spatial differences [46]. These existing studies have shown that the GWR model is more appropriate and objective than other models to explore drivers.

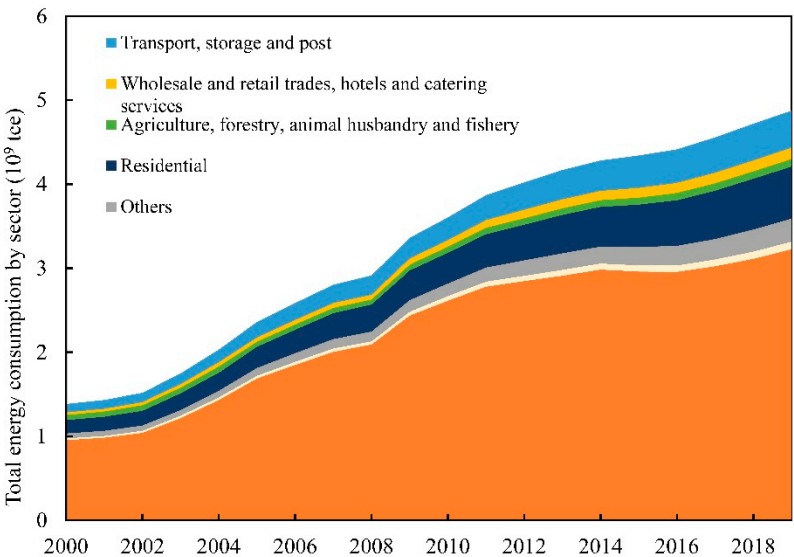

**Figure 1.** The total energy consumption by sector of China from 2000 to 2019. The industry sector is the largest energy consumption sector; the residential sector is the second largest energy consumption sector.

Given the emerging concerns about climate change, $CO_2$ emissions and energy transition are advocated and are becoming a popular research topic. Zhao et al. developed a bottom-up national energy technology model to reveal the energy consumption, $CO_2$ emissions, and technology pathways at national and sectoral levels for China's energy system's transformation and $CO_2$ emissions mitigation [47]. Wang et al. applied the decoupling index model and decomposition approach to understanding the effects of energy transition on the decoupling of economic growth from $CO_2$ emission growth [48]. Li et al. proposed the "coal–gas transition balance theory" to discuss the impact of fossil energy consumption transition on $CO_2$ emissions [49]. However, existing studies on $CO_2$ emissions reduction and energy transition are rarely conducted in the residential sector. There is also a fact that $CO_2$ emissions may spill over through climate ingredients and economic behavior [50]. The development of China's economy, interregional economic exchanges, and people's travel inevitably strengthen interregional connections. At the same time, vast territory, energy resource endowment differences, and income inequality among provinces strengthen the spatial heterogeneity of direct RCEs [51,52], which poses a challenge for

regional coordination of $CO_2$ emissions reduction and energy transition [53]. As a result, integrating interregional association and heterogeneity is important for the synergistic and differentiated energy-related $CO_2$ emissions reduction, energy transition of the residential sector, and achieving peak $CO_2$ emissions in China.

Existing research has already dealt with residential energy-related $CO_2$ emissions and energy transition, but there are still blank spaces in the research. Although SNA is used in the field of energy consumption and energy-related $CO_2$ emissions, it has rarely been applied in the study of RCEs. The existing studies on the spatial association network of energy consumption and energy-related $CO_2$ emissions considered only the association between provinces; spatial heterogeneity was ignored. In addition, $CO_2$ emissions and energy transition have received much attention as the adverse effects of climate change appeared, but the research has rarely been conducted in specific sectors, such as the residential sector. To fill that blank, we introduce SNA to complete a study of RCEs. Meanwhile, we are first to combine SNA with GWR to analyze the spatiotemporal dynamics of China's direct RCEs and the driving factors based on comprehensive interregional association and heterogeneity of direct RCEs among provinces in China, and propose policy implications of energy transition for China's residential consumption sector. Accordingly, it is expected to provide a scientific reference point for the formulation and implementation of China's energy transition and $CO_2$ emissions mitigation policies. The limitations of some existing studies and the innovation of this paper are presented in Table 1.

**Table 1.** A summary of relevant studies on $CO_2$ emissions from residential energy consumption.

| Author(s) | Methods | Subject | Area and Period | Limitations and Innovation |
|---|---|---|---|---|
| Zhou et al. [24] | Spatial autocorrelation analysis; GWR model | RCEs in China | China 2003–2015 | The attribute rather than the relationship is considered when using spatial autocorrelation analysis to study spatial effects. |
| Long et al. [25] | Spatial autocorrelation analysis; panel data | $CO_2$ emissions generated by private cars | Japan 1990–2016 | |
| Rong et al. [26] | Spatial autocorrelation analysis; GWR model | RCEs in Kaifeng | Kaifeng in China 2015–2016 | |
| He et al. [31] | SNA and the quadratic assignment procedure (QAP) | $CO_2$ emissions from the electricity sector | China 2005–2016 | SNA is a method that quantifies relationships instead of the attribute of network actors and their connections, but it has rarely been applied in the study of RCEs. |
| Shen et al. [33] | Theil index and SNA | Synergistic emissions reduction effect of urban agglomerations | Yangtze River Delta, Chengdu–Chongqing urban agglomeration, and Guangdong–Hong Kong–Macao urban agglomeration in China 2010–2015 | |
| Bu et al. [30] | SNA and LMDI model | Interprovincial natural gas consumption | China 2005–2017 | |
| Bai et al. [32] | SNA and QAP | Transportation $CO_2$ emissions | China 2005–2015 | |
| Wang et al. [38] | CLA model and STIRPAT model | Driving factors of RCEs | China 2006–2015 | These studies mainly used econometric methods that cannot investigate differences in the effects of drivers in spatial location. |
| Zang et al. [39] | LMDI model | Driving factors behind direct RCEs | Shanxi Province in China 1995–2014 | |
| Yuan et al. [42] | SDA model | Driving factors of indirect RCEs | China 2002–2007 | |

**Table 1.** *Cont.*

| Author(s) | Methods | Subject | Area and Period | Limitations and Innovation |
|---|---|---|---|---|
| Zhao et al. [47] | National energy technology model | Transition path of China's energy system | China 2015–2050 | These studies just studied the national and regional $CO_2$ emissions and energy transition, rarely conducted in the residential sector. |
| Wang et al. [48] | Tapio decoupling model and LMDI model | The effects of energy transition on the decoupling of economic growth from $CO_2$ emission growth | 186 countries in the world 1990–2014 | |
| Li et al. [49] | Coal–gas transition balance theory | The relationship between fossil energy consumption transition and $CO_2$ emissions | Sichuan Province in China 2005–2019 | |
| This paper | SNA and GWR model | RCEs and policy implications of energy transition | China 2000–2019 | This paper is the first to combine SNA and GWR to reveal the spatiotemporal dynamics characteristics and driving factors of direct RCEs, then gives some policy implications of energy transition. |

## 2. Materials and Methods

### 2.1. Data Description

Considering the data unavailability of Tibet's RCEs and the inconsistent statistical caliber of Taiwan, Hong Kong, and Macao, 30 provinces that include autonomous regions and municipalities in China are used as the research subjects in this paper; Tibet, Taiwan, Hong Kong, and Macao are not included. Due to the statistical lag, the latest data on residential energy consumption that we can obtain are for 2019. Considering the availability of data, we set 2000–2019 as the study period. We selected 2000 and 2019 as the study time nodes because the comparative results for starting and ending years are more convincing in showing the changes in study objects over the study period. All residential energy consumption data were obtained from the energy balance sheets of each province in the China Energy Statistical Yearbook (2001–2020) [54]. As data on residential energy consumption in Ningxia were missing from 2000 to 2002, we used linear regression to fit the trend of residential direct energy consumption, then extrapolated the missing energy consumption data. With reference to Kennedy et al. [55], carbon emission coefficients of all kinds of energy types are listed in Appendix A Table A1. The carbon emission coefficients of electricity in this paper are the carbon emission coefficients of China's regional power grids. They are shown in Appendix A Table A2. With reference to Jia et al. [56], the carbon emission coefficient of heat can be measured as 0.11 t-$CO_2$/GJ by the equivalent calorific value. The conversion factor of standard coal for various energy types is from the China Energy Statistical Yearbook (Appendix A Table A1). The data on GDP, per capita consumption expenditure (PCE), and population are from the China Statistical Yearbook. The geographic distance between provincial capitals is the spherical distance. To eliminate the effect of price changes, all data on GDP and PCE are converted into 2015 constant prices.

Energy-related $CO_2$ emissions are affected by economy, population, and energy consumption [57]. Economic factors include GDP, consumption level, income, per capita consumption expenditure, etc.; energy consumption structure and per capita energy consumption are energy consumption factors; population size, household size, educational level, and population aging are demographic factors [24]. GDP is an important indicator of economic development, and economic growth is the main factor of $CO_2$ emission growth [58]. Energy consumption is closely related to $CO_2$ emissions, and Reinders et al. found that energy consumption of households varies with expenditure [59]. The impact of

per capita consumption expenditure (PCE) on direct RCEs is worth exploring. Changes in per capita energy consumption were the important drivers underlying the increase in residential $CO_2$ emissions across all of Japan's 47 prefectures [60]. It should be investigated whether per capita energy consumption (PEC; the ratio of the total energy consumption to the total population) also has a great influence on direct RCEs in each province of China. The energy consumption structure, expressed by the ratio of natural gas and electricity consumption to total energy consumption, has a negative impact on direct RCEs [45]. It is important for the policy formulation of energy transition to figure out the impact of energy consumption structure (ECS), expressed by the proportion of coal consumption to total energy consumption. RCEs are influenced by consumers' lifestyles, which are reflected in consumption behavior [21]. According to Bin and Dowlatabadi, the external environment has the greatest impact on consumer behaviors, and external environment is closely related to culture and consciousness [13]. The difference in educational level (EDU; the ratio of the population with college education and above to the total population) can be a good reflection of the difference in culture and consciousness. With the population aging (AG; the ratio of the population aged 65 and above to the total population) of China deepening, it is necessary to investigate whether population aging plays a role in growth of direct RCEs or not. Given the above-mentioned contents, GDP and PCE were selected to measure economic factors; AG and EDU were chosen as demographic factors; PEC and ECS were selected as energy consumption factors.

### 2.2. Calculation of Direct RCEs

The energy sources involved in measuring the direct RCEs are raw coal, washed coal, other washed coal, briquettes, coke, coke oven gas, other gas, natural gas, liquefied natural gas, crude oil, gasoline, kerosene, diesel, lubricants, fuel oil, liquefied petroleum gas, other petroleum products, electricity, and heat. The calculation of direct RCEs uses the method provided by the IPCC [54]. The equation is:

$$C_{\mathrm{dir}} = \sum E_i \cdot F_i \tag{1}$$

where $C_{dir}$ is direct RCEs; $i$ is the type of fuel; $E_i$ is the apparent consumption of $i$; and $F_i$ represents the carbon emission factor of $i$.

### 2.3. Spatial Association Network Construction of Direct RCEs

A spatial association network of direct RCEs is an aggregate of exploring the direct RCEs relationships between provinces. Each province is a node in the network; the spatial association of direct RCEs for two different provinces is expressed by a directed line segment. There are two main methods for portraying spatial associations, namely, vector autoregressive model (VAR) and gravity model. The VAR model is only applicable to data covering a lengthy time span. Moreover, the VAR model cannot reveal the dynamic evolutionary characteristics of network structures [61]. The gravity model has many advantages for a quantitative study of spatial associations and their effects compared with the VAR model [31,33,62]. It is more applicable to aggregate data and easier to use with cross-sectional data for portraying the changing trend of spatial associations. It can also consider the comprehensive influence of various factors. Considering the difference in direct RCEs' scale between provinces, we used the share of direct RCEs of the province $i$ in the sum of direct RCEs of the provinces $i$ and $j$ to correct the empirical constant. Based on these considerations, we constructed a spatial association network of China's direct RCEs using the corrected gravity model. The equation is:

$$Y_{ij} = k \frac{\sqrt[3]{P_i G_i C_i} \, \sqrt[3]{P_j G_j C_j}}{D_{ij}^2} \tag{2}$$

$$k = \frac{C_i}{C_i + C_j} \tag{3}$$

$$D_{ij}^2 = \left[ \frac{d_{ij}}{g_i - g_j} \right]^2 \tag{4}$$

where $i$ and $j$ represent provinces $i$ and $j$, respectively; $Y_{ij}$ is the association of direct RCEs between $i$ and $j$; $d_{ij}$ is the geographic distance expressed as the spherical distance between the provincial capital city $i$ and $j$; $D_{ij}$ is the economic geographic distance between province $i$ and province $j$; $P$, $G$, $C$, and $g$, respectively, represent the total year-end population, GDP, direct RCEs, and per capita GDP; $k$ denotes the share of direct RCEs of the province $i$ in the sum of direct RCEs of the province $i$ and $j$; and $k$ is an empirical constant of the gravity model in this study.

Based on the corrected gravity model, we calculated the direct RCEs' gravitational matrix between provinces to reflect the strength of direct RCEs' gravitational force between provinces. We set the average value of each row of the gravitational matrix as the row threshold. If a value in the row is bigger than the threshold, it is designated as 1. This indicates that there is a clear spatial association of residential direct $CO_2$ emissions between the province in the row and the province in the column. Otherwise, it is designated as 0, which indicates that there is no association. According to this rule, a binary matrix can be obtained.

*2.4. Network Characteristics*

2.4.1. Overall Network Characteristics

Network density and relatedness (network connectedness, network efficiency, network hierarchy) are usually used to describe the overall network characteristics in SNA. The network density indicates how tightly connected the spatial network is, the higher the density is, the greater the association of the network is. The network density, $D$, is calculated by the following formula:

$$D = \frac{M}{N(N-1)} \tag{5}$$

where $N$ is the number of nodes; $N(N-1)$ is the maximum possible number of network relationships; and $M$ is the number of relationships.

The network connectedness represents the network structure's robustness and vulnerability. The greater the connectedness is, the more stable the network structure is. The formula is:

$$C = 1 - \frac{V}{N(N-1)/2} \tag{6}$$

where $C$ denotes connectedness; $V$ is the number of unreachable pairs of nodes in the network; and $N$ is the number of nodes.

The network hierarchy reflects the position of supremacy of network members in the network. The higher the value is, the clearer the class difference between nodes is. The network hierarchy, $H$, is calculated by the following equation:

$$H = 1 - \frac{K}{Max(K)} \tag{7}$$

where $K$ is the number of symmetrically reachable node pairs in the network; and $Max(K)$ is the maximum number of symmetrically reachable node pairs.

Network efficiency refers to the degree of redundancy of network connections. The lower the network efficiency is, the more redundant connections exist between provinces. This indicates that the network is more stable. The network efficiency, $E$, is calculated by the following formula:

$$E = 1 - \frac{R}{Max(R)} \tag{8}$$

where $R$ is the number of redundant lines in the network, and $Max(R)$ is the maximum possible redundant lines.

2.4.2. Individual Network Characteristics

Centrality is a valid index that can help us understand the importance of the roles played by individual provinces in the association network and helps to formulate regional policies accordingly. Individual network characteristics can be characterized by degree centrality and betweenness centrality. Degree centrality and betweenness centrality can be calculated directly by UCINET 6 software. Moreover, they both reflect the "power" of certain nodes in the network and the collaborative relationship between nodes. In other words, they measure how much "clout" a province has with other provinces.

The degree centrality indicates the central position of the node in the network. A node that has a higher degree centrality has tighter and more connections with other nodes, and is closer to the center of the network. The equation is:

$$De = \frac{n}{(N-1)} \tag{9}$$

where $De$ is the degree centrality; $n$ denotes the number of nodes directly associated with the node, it can be easily understood as the number of 1s in a row of the node in the binary matrix.

Betweenness centrality reflects the degree to which a node controls the association relationships of other nodes. If a node has a greater betweenness centrality, it indicates the node has a greater power to control the ties with other provinces and is closer to the center of the network. The betweenness centrality, $C_B$, is calculated by the following equation:

$$C_B = \frac{2\sum_j^n \sum_k^n b_{jk}(i)}{n^2 - 3n + 2}, j \neq k, and\ j < k \tag{10}$$

where $g_{jk}$ is the number of shortcuts that exist between nodes $j$ and $k$; $b_{jk}(i)$ denotes the ability of a third node, $i$, to control the interactions between $j$ and $k$, which is the probability that $i$ is on the shortcut between nodes $j$ and $k$; $g_{jk}(i)$ represents the number of shortcuts that exist between $j$ and $k$ through node $i$, and $b_{jk}(i) = g_{jk}(i)/g_{jk}$.

*2.5. Spatial Clustering Analysis*

The block model is the main method of spatial clustering analysis in SNA that helps us understand intuitively the role of each province in the spatial network. It divides actors in the network into discrete subsets following certain criteria and examines whether there are relationships between the subsets. These subsets are called "locations", which can also be called "clusters" and "blocks". The interaction mechanisms and influence paths among the clusters can be evaluated by a density matrix and image matrix of the network. A convergent correlations (CONCOR) method allows partitioning of each actor (province) to simplify data. The size of dataset is small in terms of the number of nodes in this study. If the dataset was divided into more than 4 clusters, there would be more clusters consisting of only a few nodes. Consequently, defective results of spatial clustering analysis will occur. Therefore, we applied a CONCOR method to perform block model analysis, and chose an iteration criterion of 0.2 and a maximum partition density of 2 to obtain four clusters.

The spatial association network can be divided into four attribute types [63], namely, net beneficiary, bidirectional spillover, net spillover, and brokers. The cluster type is determined by comparing the ratio of expected and actual internal relationships of the cluster, and combining it with the number of the total receiving and sending relation- ships of the cluster. Assuming that there are $g$ nodes in the network, where cluster $B_k$ includes $g_k$ nodes, the maximum number of possible relationships of all members in the network is $g_k(g-1)$. Moreover, if the maximum number of possible internal relationships is $g_k(g_k - 1)$, then the ratio of expected internal relationships of the cluster $B_k$ in the rational

situation is $(g_k - 1)/(g - 1)$, which is used as an indicator to determine the type of the cluster (Table 2).

**Table 2.** The division of cluster types.

| Ratio of Internal Relationships | Ratio of Accepted Relationships | |
|---|---|---|
| | $\approx 0$ | $>0$ |
| $\geq (g_k - 1)/(g - 1)$ | Bidirectional spillover cluster | Net beneficiary cluster |
| $\leq (g_k - 1)/(g - 1)$ | Net spillover cluster | Brokers cluster |

*2.6. Geographically Weighted Regression Model*

The geographically weighted regression model is an important approach to dealing with spatial heterogeneity. It embeds spatial location of data into the regression parameters and uses locally weighted least squares methods for point-by-point parameter estimation to explore the spatial variation and related drivers of the study object with a certain scale. To analyze the driving factors of direct RCEs comprehensively and thoroughly, we established a model by incorporating economic factors (GDP and PCE), demographic factors (AG and EDU), and energy consumption factors (ECS and PEC) into the regression. If there is high multi-collinearity among the explanatory variables in a multiple regression model, it will lead to inaccurate estimation. For this reason, multi-collinearity testing should be used with the factors before performing geographically weighted regression. The basic form of the regression model is:

$$y_i = \beta_0(u_i, v_i) + \sum_k \beta_k(u_i, v_i)x_{k,i} + \varepsilon_i \tag{11}$$

where $i$ denotes the spatial location point (province) $i$; $y_i$ is the explanatory value for dependent variable (direct RCEs) at location point $i$; $(u_i, v_i)$ is the longitude and latitude coordinates of $i$; $\beta_0(u_i, v_i)$ denotes the intercept parameters for location point $i$; $x_{k,i}$ is the independent variable (GDP, AG, ECS, PEC, PCE, EDU); $k$ is the number of the independent variables ($k = 1,2,3, \ldots$ , n); $\beta_k(u_i, v_i)$ is a continuous function of geographical location, which denotes the k-th coefficient of the independent variable at location point $i$, and it is obtained by a local regression estimation, which varies across the geographical location; $\varepsilon_i$ is the error of $i$.

**3. Results**

*3.1. Amount and Structure of Direct RCEs*

The amount and structure of China's direct RCEs from 2000 to 2019 are shown in Figure 2 (the amount of China's direct RCEs is calculated by Equation (1)). The total direct RCEs of China show a significant increasing trend rising from 396.32 Mt in 2000 to 1411.69 Mt in 2019. Only the amount of coal-related direct RCEs went down, whereas the direct RCEs generated by the other four energy sources went up. Specifically, coal-related direct RCEs dropped from 211.90 Mt in 2000 to 137.23 Mt in 2019, and the share decreased remarkably from 53.47% in 2000 to 9.72% in 2019. Conversely, the electricity-related direct RCEs rose most remarkably from 114.70 Mt in 2000 to 699.14 Mt in 2019 and the share increased from 28.94% in 2000 to 49.53% in 2019. The reasons for the difference in direct RCEs generated by these two kinds of energy are the popularity of household appliances and improvements in coal utilization efficiency in China. Oil-related direct RCEs also had a relatively significant upward trend, increasing from 35.11 Mt in 2000 to 249.01 Mt in 2019; the share rose from 8.86% to 17.64%. It should be noted that the share of oil-related direct RCEs was greater than the share of direct RCEs generated by coal, natural gas, and heat in 2019. The main reason is that economic development and rising incomes have produced improved living standards and demand for transport. In addition, economic development boosts demand for cleaner energy, which explains the increase in direct RCEs induced by natural gas during the investigated period. The direct RCEs generated by natural gas rose

from 7.23 Mt in 2000 to 105.15 Mt in 2019; the share grew from 1.82% to 7.45%. Heat-related direct RCEs also had an upward trend with $CO_2$ emissions growing from 27.37 Mt to 221.16 Mt; the share increased from 6.91% in 2000 to 15.67% in 2019. In summary, the amount and structure of direct RCEs varied significantly over the study period. Although the share of coal-related direct RCEs dropped remarkably, the share of oil-related direct RCEs rose. The share of direct RCEs caused by natural gas has risen, but the overall share was relatively low. Moreover, given that electricity and heat are mainly produced by coal in China, it can be seen clearly that the energy-using structure in the residential sector has improved over the investigated time, but the structure has not been altered completely. Therefore, it is necessary to promote emissions reduction and energy transition.

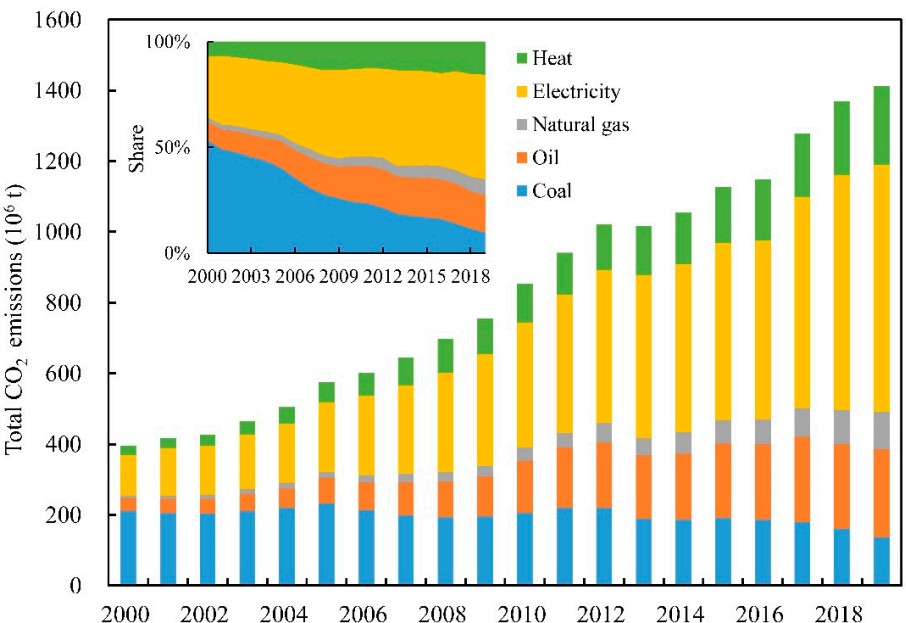

**Figure 2.** The changing trends of amount and structure of China's direct RCEs from 2000 to 2019.

### 3.2. Characteristics of Spatial Association Network

According to the corrected gravity model (Equation (2)), we calculated the spatial association matrix to establish the spatial association network of China's direct RCEs. To reveal the characteristics of the spatial association network of China's direct RCEs, we used UCINET to visualize the network (Figure 3).

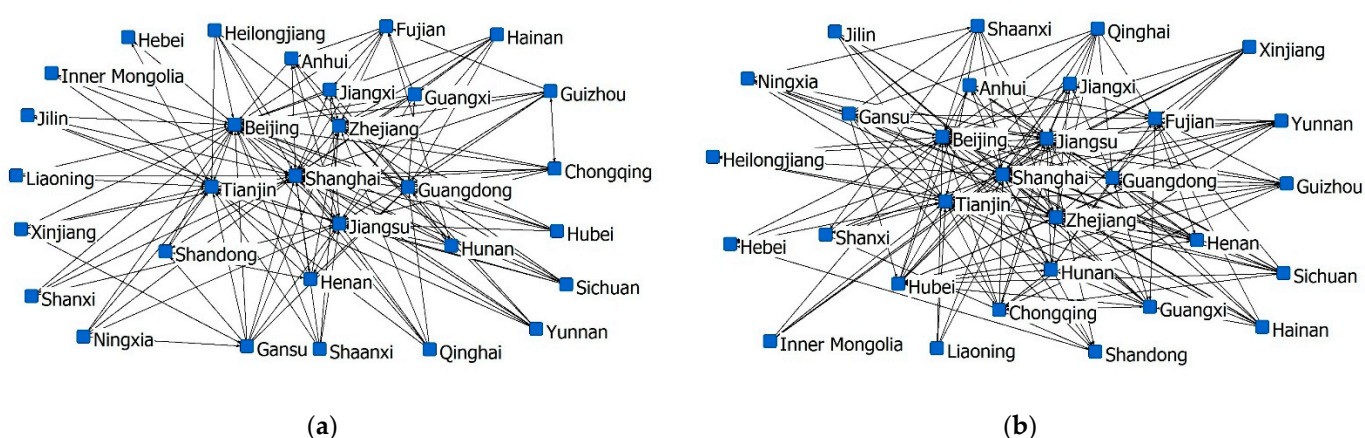

(**a**)                              (**b**)

**Figure 3.** (**a**) The spatial association network of China's direct RCEs in 2000; (**b**) the spatial association network of China's direct RCEs in 2019.

### 3.2.1. Overall Network Characteristics

The spatial associations of China's direct RCEs had a typical network structure and an obvious center–edge structural feature. The number of associations of direct RCEs has increased over the two decades (Figure 3). In this study, network density, network connectedness, network efficiency, and network hierarchy are selected to describe the overall network characteristics, they are calculated by Equations (5)–(8), respectively.

The number of network association relationships and the network density of China's direct RCEs both present a trend of fluctuating growth. The number of network association relationships increased from 150 in 2000 to 192 in 2019. The network density rose from 0.1724 in 2000 to 0.2207 in 2019. The network connectedness always had a value of 1 over the investigated period. This indicates that direct RCEs in China had universal direct or indirect associations, and the interprovincial spatial associations were becoming stronger. However, the maximum number of association relationships, 192 in 2019, shows a large gap from the maximum possible number of association relationships of 870 (Figure 4). This means the spatial association network of China's direct RCEs is continuously optimizing, but it is far from the best state.

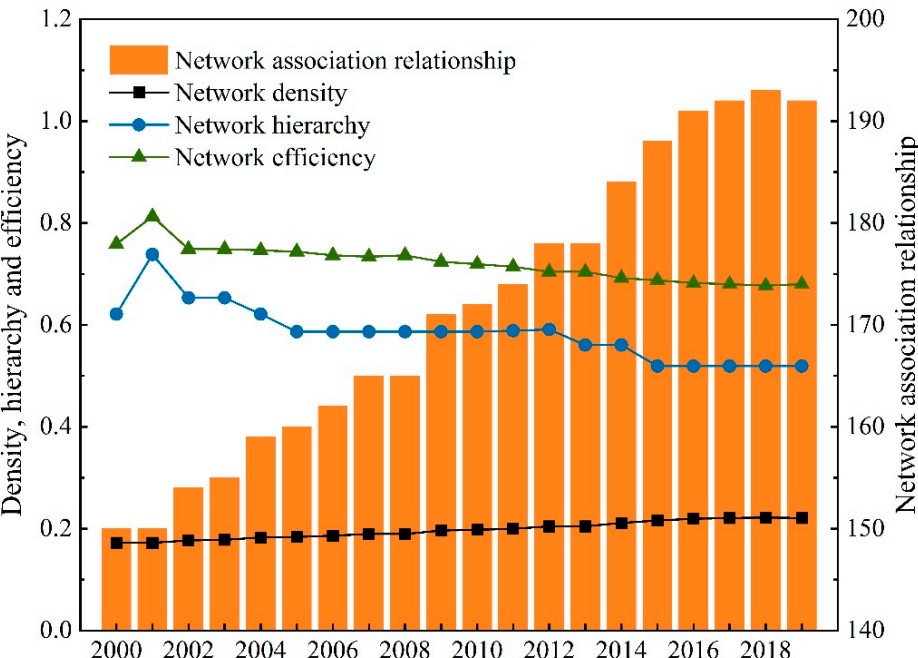

**Figure 4.** The density, hierarchy, efficiency, and association relationships of the spatial association network for China's direct RCEs from 2000 to 2019.

The network hierarchy and efficiency both had a downward trend. The network hierarchy decreased from 0.6212 in 2000 to 0.519 in 2019, but it is still greater than 0.5 (Figure 4). This indicates that the interprovincial spatial associations of direct RCEs have been strengthened. Although the hierarchical structure of the network is gradually being broken, the network still has a strong spatial hierarchy. The network efficiency decreased from 0.7586 in 2000 to 0.6798 in 2019. Although the network efficiency increased from 2000 to 2001, there was a decline in other years and a small fluctuation in the overall trend. This further confirms that the spatial associations of direct RCEs became stronger.

### 3.2.2. Individual Network Characteristics

Individual network characteristics can be analyzed by degree centrality and betweenness centrality. In this study, we used Equations (9) and (10) to calculate degree centrality and betweenness centrality. The results in 2000 and 2019 are shown in Figure 5 and Appendix A Table A3. The average degree centrality was 29.20 in 2000 and 36.55 in 2019. The mean values of betweenness centrality in 2000 and 2019 were 2.55 and 2.29, respectively.

Beijing, Shanghai, and Jiangsu were the top three provinces; their degree centrality and betweenness centrality were greater than average in 2000 and 2019. Beijing, Shanghai, and Jiangsu are economically developed regions in China. They had a higher level of economic development and transport. For these reasons, they were at the center of the spatial association network for China's direct RCEs and had a significant influence on the overall spatial associations for direct RCEs. It must be noted that they control the spatial associations of other provinces in the network. Conversely, provinces such as Inner Mongolia, Liaoning, Hebei, Xinjiang, Ningxia, and Jilin were at the edge of the spatial association network for China's direct RCEs with a lower degree centrality and betweenness centrality. They are concentrated in remoter geographical areas of northwest and northeast China. These provinces had a relatively small economic development scale. Therefore, their associations were more easily dominated by the provinces that occupy a central position in the spatial network.

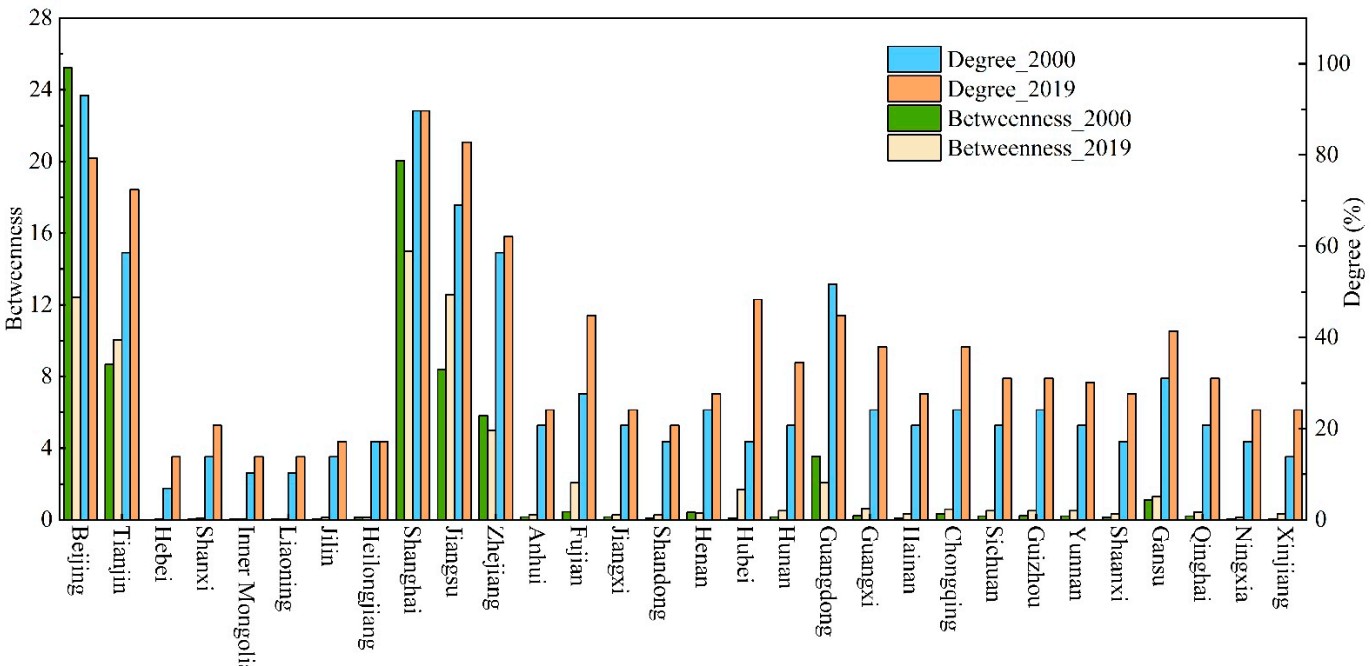

**Figure 5.** The degree centrality and betweenness centrality of the spatial association network for China's direct RCEs in 2000 and 2019. The detailed data are shown in Appendix A Table A3.

### 3.3. Spatial Clustering Analysis

The results of spatial clustering analysis in 2000 and 2019 are shown in Figure 6 and Table 3. In 2000, cluster A contained 14 provinces; cluster B included 10 provinces; clusters C and D both contained three provinces. The spatial association network had 150 associations with 12 internal relationships and 138 external relationships. The ratio of expected internal relationships of cluster A was greater than the ratio of actual internal relationships. Furthermore, the number of contacts received from outside was significantly lower than the number of contacts sent to the outside. Therefore, cluster A was the net spillover cluster. The ratio of expected internal relationships of cluster B was greater than the ratio of actual internal relationships, and the number of relationships sent was significantly larger than the number of relationships received. Thus, cluster B is also a net spillover cluster. The ratio of actual internal relationships of cluster C was smaller than the ratio of expected internal relationships. Cluster C not only sent relationships to the outside but also received relationships from the outside. The relationships received from the outside were more than those sent to the outside, and the internal relationships of cluster C account for a lower proportion. Therefore, cluster C was the brokers cluster and was an intermediary in the network. The ratio of actual internal relationships of cluster D

was greater than the ratio of expected internal relationships, and the relationships received from the outside were much greater than the relationships sent to the outside. Thus, cluster D was the net beneficial cluster.

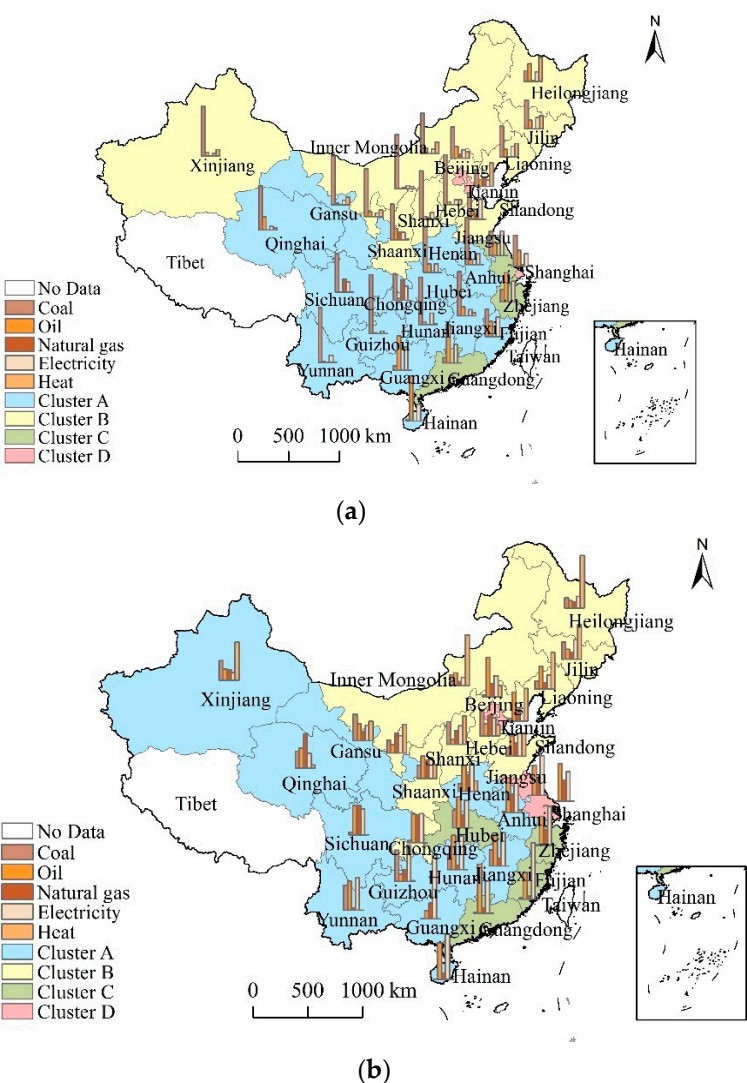

**Figure 6.** (**a**) The clusters' spatial distribution of the spatial association network for China's direct RCEs in 2000; (**b**) the clusters' spatial distribution of the spatial association network for China's direct RCEs in 2019.

**Table 3.** The clusters' spillover effect of spatial association network for China's direct RCEs in 2000 and 2019.

| Clusters | | Receiving Relationships | | Sending Relationships | | Ratio of Expected Internal Relationships | Ratio of Actual Internal Relationships [a] | Cluster Type |
|---|---|---|---|---|---|---|---|---|
| | | Inside | Outside | Inside | Outside | | | |
| 2000 | A | 10 | 17 | 10 | 74 | 44.83% | 11.90% | Net Spillover |
| | B | 0 | 6 | 0 | 39 | 31.03% | 0 | Net Spillover |
| | C | 0 | 47 | 0 | 16 | 6.90% | 0 | Brokers |
| | D | 2 | 68 | 2 | 9 | 6.90% | 18.18% | Net Beneficial |

Table 3. *Cont.*

| Clusters | | Receiving Relationships | | Sending Relationships | | Ratio of Expected Internal Relationships | Ratio of Actual Internal Relationships [a] | Cluster Type |
|---|---|---|---|---|---|---|---|---|
| | | Inside | Outside | Inside | Outside | | | |
| 2019 | A | 3 | 32 | 3 | 92 | 37.93% | 3.16% | Net Spillover |
| | B | 4 | 14 | 4 | 48 | 31.03% | 6.00% | Net Spillover |
| | C | 1 | 45 | 1 | 26 | 10.34% | 3.70% | Brokers |
| | D | 2 | 91 | 2 | 16 | 10.34% | 11.11% | Net Beneficial |

[a] The ratio of the clusters' numbers of internal relationships to the clusters' total numbers of spillover relationships.

In 2019, cluster A contained 12 provinces; cluster B included 10 provinces; clusters C and D both contained four provinces. The spatial association network of China's direct RCEs had 192 associations with 10 internal relationships and 182 external relationships. The ratio of expected internal relationships is 37.93%; the ratio of actual internal relationships is 3.16%. Obviously, the ratio of expected internal relationships was greater than the ratio of actual internal relationships. Furthermore, the number of relationships received from the outside was significantly lower than the number of relationships sent to the outside. Therefore, cluster A was the net spillover cluster. It should be noted that cluster B was defined as the net spillover cluster for the same reasons as cluster A. The ratio of actual internal relationships of cluster C was smaller than the ratio of expected internal relationships. The relationships received from the outside were more than those sent to the outside, and the internal relationships of cluster C account for a lower proportion. Therefore, cluster C was the brokers cluster in the network. Cluster D was the net beneficial cluster, because the ratio of actual internal relationships was greater than the ratio of expected internal relationships, and the relationships received from the outside were much greater than the relationships sent to the outside.

To sum up, in 2000 and 2019, clusters A and B were the net spillover cluster; cluster C was the brokers cluster; cluster D was the net beneficial cluster. The provinces that were classified as clusters A, B, C, and D between 2000 and 2019 just changed slightly. The provinces of cluster A were undeveloped and mostly had a higher proportion of coal energy consumption than other clusters. These provinces were in central and western China. The provinces of cluster B were mainly resource-rich but economically underdeveloped. These provinces are in central, northern, and northeast China. Most provinces of cluster D were in developed eastern China and had the lowest proportion of coal energy consumption. It should be noted that the number of provinces that were classified as cluster A in 2019 is less than that in 2000. Fujian and Hubei were not classified as the net spillover cluster (cluster A) in 2019, but classified as the brokers cluster (cluster C). This change can be explained by the following reasons. First, in 2000, the eastern coastal area was the center of China's economic development according to the strategy of preferential development of the east. Since the implementation of the Central Rising Development Strategy in 2004, Hubei's economy and infrastructure have been developing rapidly. Wuhan, the capital of Hubei, has become the economic and geographical center of China by being a "bridge" between the energy-rich area in western China and the economically developed area in the Yangtze River Delta. Secondly, since the construction of the economic zone on the west side of the strait in 2019, Fujian has become increasingly connected to both the Pearl River Delta and the Yangtze River Delta economic zones. Since Jiangsu implemented the Yangtze River Delta Regional Integration Strategy in 2010 and the Yangtze River Economic Belt Development Strategy in 2016, the comprehensive development level of Jiangsu has been significantly improved. Now, it is one of the provinces with the highest level of comprehensive development in China. This is the reason Jiangsu was classified as the brokers cluster (cluster C) in 2000 and changed into a province of the net beneficial cluster (cluster D).

We calculated the density matrix to reflect the spillover effect among the clusters, and examined the transmission mechanism of the direct RCE flow among provinces. The

density of the spatial association network for China's direct RCEs was 0.1427 in 2000 and 0.2207 in 2019, which were taken as the critical values. If the density of each cluster was greater than the critical value, a value of 1 was assigned, otherwise, a value of 0 was assigned. According to this rule, the image matrix was obtained (Table 4). The rows of the image matrix represent the sending relationships; the columns denote the receiving relationships; the diagonal lines represent the associations among the members within the cluster.

**Table 4.** The density and image matrix of the spatial association network for China's direct RCEs in 2000 and 2019.

| Clusters | | Density Matrix | | | | Image Matrix | | | |
|---|---|---|---|---|---|---|---|---|---|
| | | A | B | C | D | A | B | C | D |
| 2000 | A | 0.055 | 0.014 | 0.905 | 0.81 | 0 | 0 | 1 | 1 |
| | B | 0.021 | 0 | 0.233 | 0.967 | 0 | 0 | 0 | 1 |
| | C | 0.262 | 0 | 0 | 0.556 | 0 | 0 | 0 | 1 |
| | D | 0.071 | 0.133 | 0.222 | 0.333 | 0 | 0 | 1 | 1 |
| 2019 | A | 0.023 | 0.058 | 0.813 | 0.958 | 0 | 0 | 1 | 1 |
| | B | 0.050 | 0.044 | 0.100 | 0.95 | 0 | 0 | 0 | 1 |
| | C | 0.396 | 0.000 | 0.083 | 0.438 | 1 | 0 | 0 | 1 |
| | D | 0.146 | 0.175 | 0.125 | 0.167 | 0 | 0 | 0 | 0 |

In 2000 and 2019, clusters A and B were the primary emitters of spillover relationships; clusters A, B, and C sent relationships to other clusters; cluster D was the primary recipient of spillover relationships and was associated with other clusters (Table 4). Clusters A and B could deliver energy to the eastern economic development areas (clusters C and D) as they had abundant coal, oil, and natural gas resources. Additionally, clusters A and B had a higher proportion of coal consumption. Therefore, clusters A and B were important areas that can develop clean energy and promote energy consumption structure transition. That cluster C played an intermediary role in the network in 2000 and 2019 can be explained by two points. First, Zhejiang is in the Yangtze River Delta, which has frequent economic activity transactions and close population exchanges with Shanghai and Jiangsu. Secondly, Guangdong has close economic ties with the Yangtze River Delta and the Beijing–Tianjin–Hebei area. The provinces of cluster D are the economically developed Chinese provinces with a relative lack of energy resources. Therefore, they need to receive a lot of energy from other clusters to meet the needs of their economic development. These provinces have a siphon effect on the other areas and the various elements such as population, technology, and resource flow to cluster D. Therefore, cluster D is the core, final link of the spatial association network of China's direct RCEs; it is the priority area for energy transition and $CO_2$ reduction in China.

### 3.4. Driving Factors' Analysis

The six driving factors were standardized (dimensionless) and were tested by the variance inflation factor (VIF) before performing regression analysis. The results show that there is no multi-collinearity among the factors (VIF ≤ 7.5), and regression analysis can be performed (Table 5).

**Table 5.** The multi-collinearity diagnosis results of driving factors for China's direct RCEs in 2000 and 2019.

| VIF | GDP | AG | ECS | PEC | PCE | EDU |
|---|---|---|---|---|---|---|
| 2000 | 1.679 | 2.391 | 1.602 | 2.899 | 5.502 | 5.046 |
| 2019 | 1.670 | 1.168 | 1.716 | 1.778 | 5.634 | 5.673 |

Table 6 presents the overall estimates of the GWR model and the average regression coefficient of each factor for 30 provinces in 2000 and 2019, which were calculated by Equation (10) using ArcGIS 10.2 software. It can be seen from the estimates that the overall fit of the GWR model was relatively good. From the perspective of coefficients' average values, the coefficients of GDP, PEC, PCE, and AG were the top four from 2000 to 2019. The ranking of the regression coefficient for each factor changed slightly in 2019 compared with 2000. Specifically, the regression coefficient of EDU changed from fifth to sixth, and the regression coefficient of ECS changed from sixth to fifth. In other words, the influence of EDU was weaker than ECS in 2019.

**Table 6.** The overall estimates of the geographically weighted regression model and the average regression coefficient of each factor for 30 provinces in 2000 and 2019.

| | Overall Estimates | | | | Average Regression Coefficients | | | | | |
|---|---|---|---|---|---|---|---|---|---|---|
| | $R^2$ | Adjust $R^2$ | Bandwidth | $AIC_C$ | GDP | AG | ECS | PEC | PCE | EDU |
| 2000 | 0.762 | 0.666 | 2789137.739 | 472.742 | 0.893 | 0.247 | 0.121 | 0.794 | −0.702 | −0.134 |
| 2019 | 0.798 | 0.745 | 35816131.810 | 540.503 | 0.941 | 0.192 | 0.104 | 0.439 | −0.381 | −0.011 |

To determine the variation of the regression coefficient of each factor across provinces, the regression coefficient for each province is presented in Figure 7. There is an upward trend for every province for the influence of GDP, but the influence of other factors decreased with increasing years. GDP, AG, ECS, and PEC had positive contributions to the growth of each province's direct RCEs, whereas PCE and EDU had negative impacts. GDP, PEC, and PCE had significant influences on the growth of direct RCEs, which indicates they had stronger explanatory power on direct RCEs. Therefore, the influences of GDP, PEC, and PCE deserve special attention. It should be noted that GDP and PCE are economic factors and GDP had the greatest influence. Although demographic factors (AG and ECS) had insignificant influence, the influence of AG was greater than EDU. Therefore, considering the influence of economy, energy consumption, and demography on direct RCEs, we focus on analyzing the spatial and temporal evolutions of the influences of GDP, PEC, and AG. We present their regression coefficients on the map of China using ArcGIS software (Figures 8–10).

The GDP had a positive contribution to the growth of direct RCEs and, except for Xinjiang, the contribution for each province's direct RCEs increased slightly over the investigated period (Figure 8). The primary reason is that energy demand increases with economic development, but energy utilization technology and environmental awareness of residents also improve. Significantly, this phenomenon is more pronounced in areas with a higher level of economic development. The positive influence of GDP displays a significant geographic variation and the degree of influence on direct RCEs was gradually enhanced from the east coastal area of China to western China from 2000–2019. The eastern coastal area of China has a higher level of economic development and energy utilization efficiency. Although the economic growth rate of western China has accelerated since the Western Development Strategy was implemented, there was a higher proportion of coal consumption in western China and less consistent economic development. Therefore, with the combination of the above reasons, the influential effect of GDP was gradually enhanced from the eastern coastal area of China to western China.

The positive influence of PEC was significant, but decreased during the investigated period (Figure 9). This is related to improvements in energy utilization efficiency in China. The spatial distribution for the positive influence of PEC in each province from 2000 to 2019 varied widely. In 2000, the greatest positive impact was in the eastern coastal area of China. However, in 2019, the greatest positive impact was in northeast China; southwest and northwest China had a smaller positive impact. Although the eastern coastal area has been the key region of China's economic development since the reform and opening up, the development pattern has not been transformed yet and awareness of energy saving

among residents was lacking in 2000. This led to the fact that the eastern coastal area of China is the region with the strongest positive influence in 2000. Southwest and northwest China have responded to national policies that positively promote clean energy production and energy consumption structure transition, so have achieved significant results. For this reason, southwest and northwest China had a smaller positive impact of PEC in 2019. Additionally, northeast China is China's heavy industrial base and its industrial structure and energy consumption structure cannot be fundamentally adjusted in a short time. Thus, in 2019, the greatest positive impact of PEC was in northeast China.

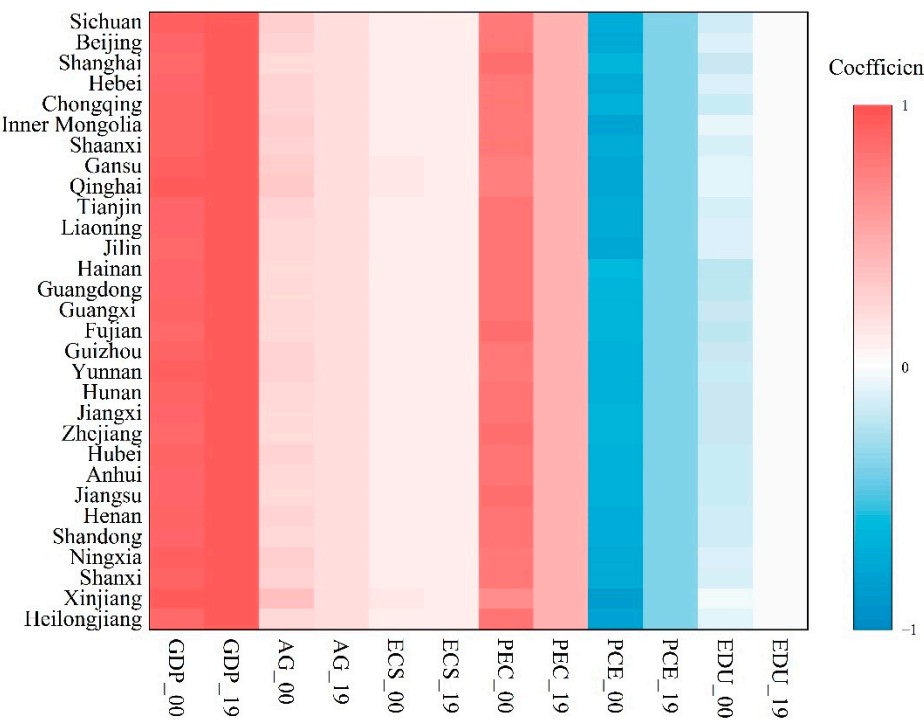

**Figure 7.** The spatiotemporal variation of the regression coefficients of the six factors across provinces in 2000 and 2019. Red represents a positive influence and blue represents a negative influence. The darker the color is, the higher the degree of the influence is.

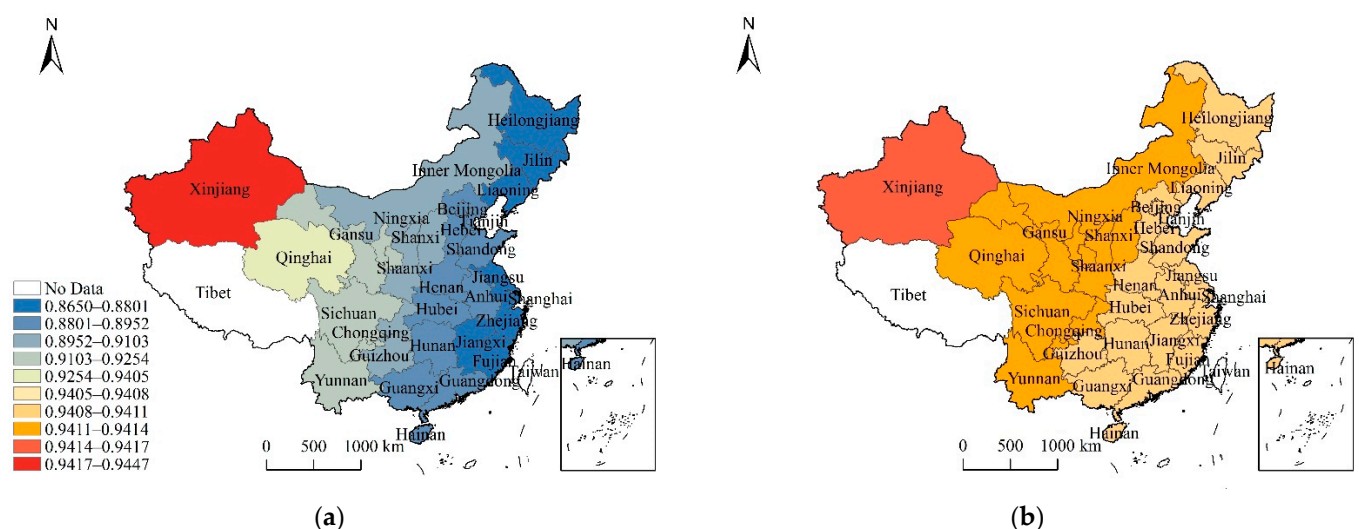

**Figure 8.** (**a**) The regression coefficients of the GDP effect in 2000; (**b**) the regression coefficients of the GDP effect in 2019.

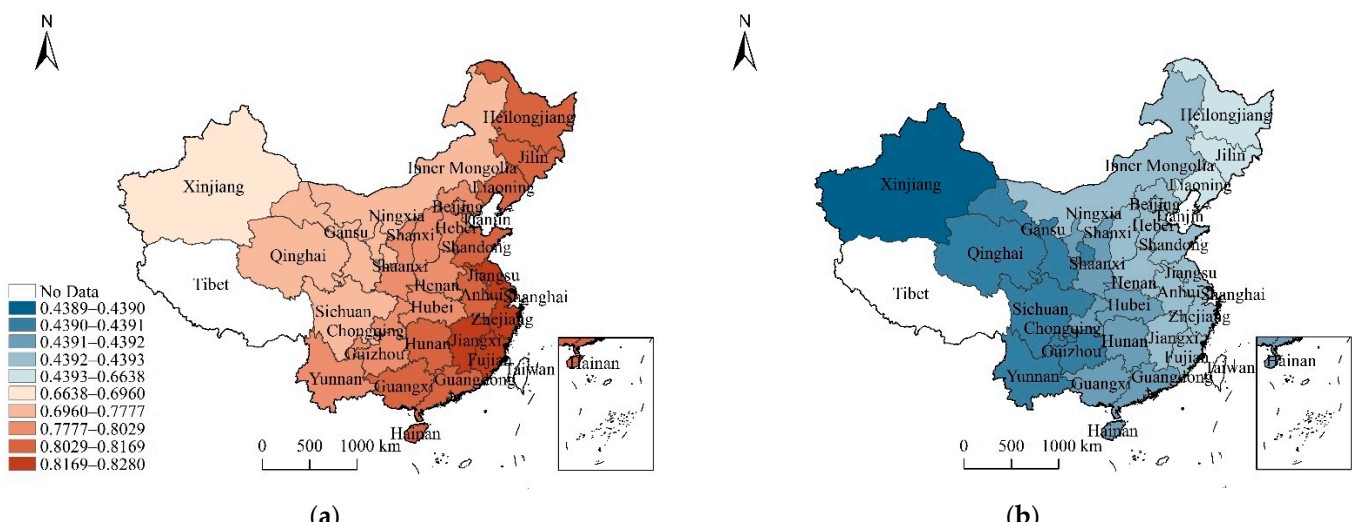

**Figure 9.** (**a**) The regression coefficients of the PEC effect in 2000; (**b**) the regression coefficients of the PEC effect in 2019.

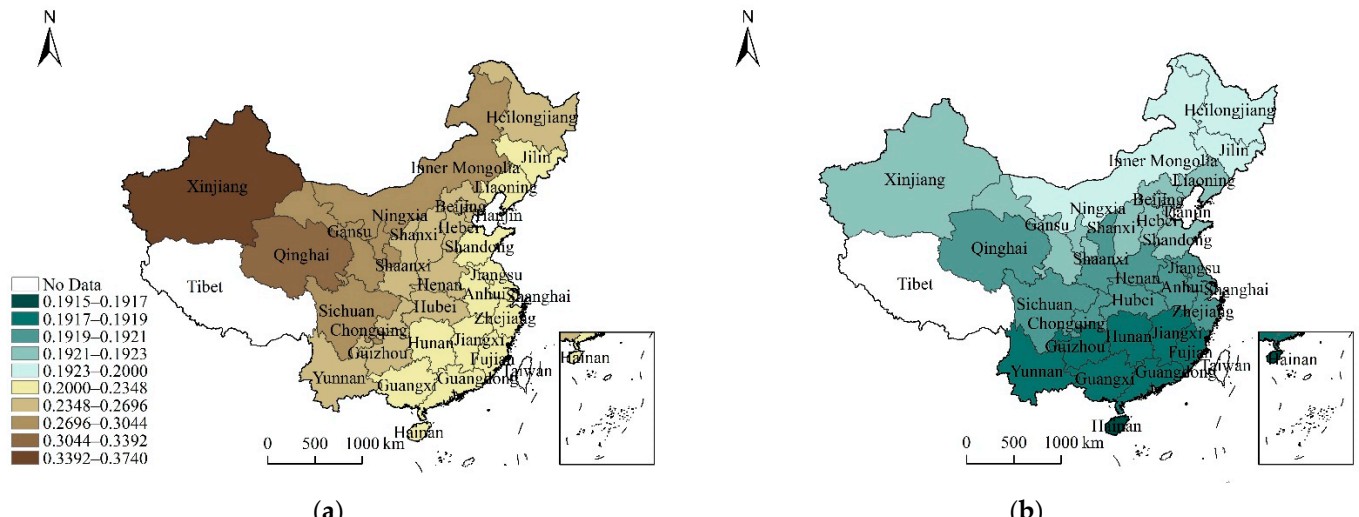

**Figure 10.** (**a**) The regression coefficient of the AG effect in 2000; (**b**) the regression coefficients of the AG in 2019.

AG had a positive impact on direct RCEs, but the positive impact was insignificant and weakening (Figure 10). The main reason is that population aging deepening and popularization of electrified equipment had led to elderly people using electrified equipment more frequently. However, the elderly Chinese are inclined to travel on foot or by public transport. They also have a strong sense of thrift and frugality, which is consistent with the reality of China. The impact of AG shows a large spatial distribution difference between 2000 and 2019. The degree of influence gradually increased from the southeast coastal area of China to northwest China in 2000. However, in 2019, AG in northeast China had the greatest impact on direct RCEs, which resulted from various factors such as serious population aging and irrational industrial structure in northeast China.

## 4. Discussion

It would be meaningful to compare the results of our study with previous studies. The spatial associations of direct RCEs show an obvious spatial network structure and the number of associations increased during the investigated period. Beijing, Shanghai, and Jiangsu were at the center of the spatial association network for China's direct RCEs.

The network characteristics of the spatial association network for China's direct RCEs are similar to the network characteristics for China's $CO_2$ emissions [64].

Currently, some advanced concepts of social network analysis have already been proposed and used to address some issues in other fields. These advanced concepts include a multi-dimensional social network-based model [65], the concept of scope in a multi-IoT scenario [66], trust and reputation in a multi-IoT scenario [67], colored social networks [68], and a social network analysis-based classifier (SNAc) [69]. Ucer et al. developed SNAc, which allows building the model based on the correlation between samples [69]. Corradini et al. used a multi-dimensional social network-based model to investigate negative reviews from multiple dimensions [65]. In our study, we give some policy implications for energy transition based on results. It is important to study whether these policy implications are widely accepted by the public. Therefore, a multi-dimensional social network-based model may be applied to study the public's positive and negative feedback on these policy implications on social platforms. In a multi-IoT scenario, there are some analogies between scope and some other concepts used in sociology such as reputation, trust, centrality, and power. The concepts of centrality and power were applied in our study. In a multi-IoT scenario, Cauteruccio et al. defined the concept of scope, and proposed two formalizations of this concept, allowing them to calculate its values [66]; Ursino and Virgili proposed a context-aware approach that is not limited to certain specific scenarios to evaluate trust and reputation of things [67]. Chen proposed a novel trust inference model to assess the trustworthiness of mobile social networks. This model is a multi-dimensional fuzzy trust inferring approach introducing a multi-dimensional trust metric to reflect the complexity of trust. Moreover, this model provides more detailed analysis and higher accuracy in trust assessment [70]. In a social context, investigating trust and reputation of things is useful, and the evaluation of service quality is one of its applications [67]. Carbon inclusion is a new direction for reducing $CO_2$ emissions in the residential sector of China. The Carbon Inclusion Platform and Carbon Inclusion Cooperation Network are designed for this direction. Combining the Carbon Inclusion platform with the Internet of Things (IoT) to stimulate reduction behaviors of residents should be considered in our future work. In addition, in a multi-IoT scenario, the scope of services, the quality of services, the reputation, and trust in the Carbon Inclusion Platform and Carbon Inclusion Cooperation Network should be evaluated by a suitable approach. Lo Giudice et al. proposed a network analysis-based approach to support experts in their analyses of subjects with mild cognitive impairment and Alzheimer's disease [68]. The important tool used in this approach is the concept of a clique. Given that the concept of a clique has not been applied to study association networks of $CO_2$ emissions, it may be considered in our future work provided that the conditions for its use are met.

We applied the GWR model to analyze the driving factors of direct RCEs. It can be found that per capita consumption expenditure had a greater influence on direct RCEs, which is similar to the finding of Zha et al. [71]. Rong et al. also found that per capita consumption expenditure was one of the main drivers of the increment in direct RCEs in central China [72]. In addition, the effect of energy consumption structure on direct RCEs was negative in the findings of Wang et al. [45], but the influence of energy consumption structure is positive in our study. The reason is that there is a different measure of energy consumption structure between the two studies. We found that the per capita energy consumption had a large positive effect on direct carbon emissions, but the impact on each province was different. Yosuke et al. came to a similar conclusion in their study of sub-national $CO_2$ emissions in Japan's household sector [60]. However, there are some limitations in this study with using a GWR model to analyze the driving factors of direct RCEs. First, direct RCEs may be influenced by the number of households [38], per capita household income [39], energy intensity [38], population size, and urbanization [37]. However, this study just analyzed six drivers: gross national product (GDP), per capita consumption expenditure (PCE), energy consumption structure (ECS), per capita energy consumption (PEC), population aging (AG), and educational level (EDU). Moreover, this

study only explored the driving factors influencing direct RCEs but did not pay any attention to studying the influencing mechanism of the role that each cluster plays in the spatial association network of direct RCEs. In addition, GWR extends the traditional linear regression framework by allowing local rather than global parameters [73]. GWR also faces a potential endogeneity issue (occurring when an independent variable is correlated with the error term) of a traditional linear regression framework [74]. However, in our study, we did not pay attention to discussing the endogeneity that probably exists. Meanwhile, most researchers have not discussed potential endogeneity when applying GWR to address the issue of driving factors in the field of energy consumption and $CO_2$ emissions, such as Chen et al. [75], Sheng et al. [44], Qin et al. [76], and Yang et al. [77]. Thus, this limitation should be given greater attention in the future.

## 5. Conclusions and Suggestions

### 5.1. Conclusions

Taking 30 provinces of China as an example, we analyzed the spatiotemporal evolution characteristics and driving factors of the direct RCEs. The conclusions are set out below.

From 2000 to 2019, the total direct RCEs rose from 396.32 Mt to 1411.69 Mt. The coal consumption and electricity consumption were the primary sources of direct RCEs. Only the amount and share of coal-related direct RCEs went down; the direct RCEs generated by the other four energy sources increased.

From 2000 to 2019, the spatial associations of direct RCEs in China show an obvious spatial network structure. The number of associations was increasing, but it still had strong spatial hierarchical characteristics. Beijing, Shanghai, and Jiangsu were at the center of the spatial network and were classified as the net beneficiary cluster (cluster D) in 2019. These provinces had a significant influence on the spatial associations. Provinces such as Yunnan, Shanxi, Xinjiang, Gansu, Qinghai, and Guizhou were classified as the net spillover cluster (clusters A and B). Zhejiang, Guangdong, Fujian, and Hubei were classified as the brokers cluster (cluster C) in 2019. The role of each cluster was different in the network. Clusters A and B were senders of direct RCE association relationships. Thus, they were facing two important issues, deliver clean energy to other clusters and promote the transition of energy consumption structure. Cluster C played the role of a "bridge" in the network. Cluster D was the core and final link of the spatial network. Thus, cluster D should be the critical and priority area of residential $CO_2$ emissions reduction and energy transition in China.

GDP, PCE, and PEC had a significant influence on the growth of direct RCEs, and AG, ECS, and EDU had an insignificant influence. GDP, AG, ECS, and PEC had a positive influence on the growth of direct RCEs, whereas PCE and EDU had a negative impact. In 2019, from the perspective of coefficients' absolute values, the influence levels of the six driving factors on the direct RCEs were: GDP (0.941), PEC (0.439), PCE (−0.381), AG (0.192), ECS (0.104), EDU (−0.011).

### 5.2. Suggestions

Based on the above conclusions, suggestions for China's emissions reduction and energy transition are proposed as follows.

First, coal consumption substitution and transition should be promoted. Coal consumption and electricity consumption were the primary sources of direct RCEs. This indicates that promoting the substitution of coal and electricity consumption is an effective scientific way to realize emissions reduction and energy transition. In other words, coal consumption should be strictly and reasonably controlled, and the use of clean coal should be accelerated. New coal power projects should be controlled and electricity generated by renewable energy should be promoted.

Secondly, the economic development pattern should be modified to be greener and be in the direction of greater sustainability and better health; a favorable social environment for low-carbon energy transition and emissions reduction should be created. Northwest China where the GDP contributed most to the growth of direct RCEs and the area where

clusters A and B were should give special consideration to the above measures. Beijing, Shanghai, and Jiangsu were at the center of the direct RCE spatial network and were classified as cluster D. They should use their central position in the network to take a lead in accelerating a comprehensive green transition of economy and society, and play the role of a growth pole for high-quality development, and drive economic development and industrial structure upgrading in surrounding areas. Hubei, Zhejiang, Fujian, and Guangdong should strengthen the efficiency of energy transmission on the premise of promoting economic development of the region.

Thirdly, the shift of residential consumption patterns to be greener for energy transition in terms of the residential sector's consumption structure should be advocated; improving energy utilization efficiency and stimulating the utilization of clean energy are necessary. On one hand, the proportion of coal consumption should be reduced for an increasing proportion of natural gas and other clean energy supplies. This measure should be implemented with special consideration in the area where clusters A and B were located and in northeastern China where PEC had the greatest influence in 2019. On the other hand, each province ought to develop clean energy based on its real conditions. Specifically, renewable energy such as solar energy and wind energy should be vigorously developed in western China, thereby continuously exporting clean energy to Shanghai, Beijing, Tianjin, Jiangsu, and other provinces with a relative lack of resources but which are highly developed. As the core, final link of the spatial association network for China's direct RCEs, Shanghai, Beijing, Tianjin, and Jiangsu should improve the utilization efficiency of carbon-containing resources and advocate for the shift of consumption patterns of residents to be greener. These provinces should strengthen their demonstration role of $CO_2$ emissions reduction and transition for energy consumption structure.

Finally, education should be developed to enhance the awareness of a low-carbon lifestyle and provide a universal foundation for emissions reduction and energy transition. Demographic factors had a certain impact on the direct RCEs. Improved education is conducive to enhancing awareness of energy saving and emissions reduction. China has implemented a three-child policy in response to the adverse socio-economic impact of population aging. The number of newborns in China will increase as a result. Given that, making greater efforts to develop education is particularly important for energy transition for now and the future. It should be noted that the impact of AG is strongest in northeast China. Thus, enhancing the awareness of a low-carbon lifestyle and encouraging people to use greener products might be an effective method for emissions reduction and energy transition in northeast China.

**Author Contributions:** Conceptualization, Y.S. and J.J.; investigation, Y.S. and J.J.; data curation, M.J. and C.C.; methodology, M.J. and C.C.; writing—original draft preparation, Y.S.; writing—review and editing, J.J. and C.C. All authors have read and agreed to the published version of the manuscript.

**Funding:** This research was funded by the Foundation Project of Philosophy and Social Science in Jiangxi Province (Grant No. 21JL03), the Research Project of Humanities and Social Science in Jiangxi's Universities (Grant No. GL19225), and the Chinese National Science Foundation (Grant No. 71473113).

**Institutional Review Board Statement:** Not applicable.

**Informed Consent Statement:** Not applicable.

**Data Availability Statement:** The data presented in this study are available on request from the corresponding author.

**Acknowledgments:** We thank the anonymous reviewers and editors for their constructive comments and suggestions to improve the quality of this article.

**Conflicts of Interest:** The authors declare no conflict of interest. The funders had no role in the design of the study; in the collection, analyses, or interpretation of data; in the writing of the manuscript, or in the decision to publish the results.

## Appendix A

**Table A1.** The carbon emission coefficients and standard coal coefficients of each energy type.

| Energy Type | Emission Coefficient (t-$CO_2$/t) | Standard Coal Coefficient (kg ce/kg) |
|---|---|---|
| Raw coal | 1.977 | 0.7143 |
| Washed coal | 2.488 | 0.9000 |
| Other washed coal | 0.795 | 0.2857 |
| Briquettes | 1.717 | 0.6000 |
| Coke | 3.019 | 0.9714 |
| Coke oven gases | 7.421 [a] | 0.6143 [b] |
| Other gases | 7.421 [a] | 0.3570 [b] |
| Natural gas | 21.84 [a] | 1.3300 [b] |
| Liquefied natural gas | 2.836 | 1.7572 |
| Crude oil | 3.102 | 1.4286 |
| Gasoline | 3.185 | 1.4714 |
| Kerosene | 3.153 | 1.4714 |
| Diesel | 3.185 | 1.4571 |
| Lubricants | 2.948 | 1.4143 |
| Fuel oil | 3.126 | 1.4286 |
| Liquefied petroleum gas | 2.983 | 1.7143 |
| Other petroleum products | 2.948 | 1.2000 |

[a] The unit is t-$CO_2$/$10^4$ $m^3$. [b] The unit is kg ce/$m^3$.

**Table A2.** The carbon emission coefficients of China's regional grids.

| Province | Emission Coefficient (kg/kw·h) | Grid Affiliation | Region | Emission Coefficient (kg/kw·h) | Grid Affiliation |
|---|---|---|---|---|---|
| Beijing | 0.8292 | NCPG | Tianjin | 0.8733 | NCPG |
| Shanxi | 0.8798 | NCPG | Inner Mongolia | 0.8503 | NCPG/NECPG |
| Jilin | 0.6787 | NECPG | Heilongjiang | 0.8158 | NECPG |
| Jiangsu | 0.7356 | ECPG | Zhejiang | 0.6822 | ECPG |
| Fujian | 0.5439 | ECPG | Jiangxi | 0.7635 | CCPG |
| Henan | 0.8444 | CCPG | Hubei | 0.3717 | CCPG |
| Guangdong | 0.6379 | SCPG | Guangxi | 0.4821 | SCPG |
| Chongqing | 0.6294 | CCPG | Sichuan | 0.2891 | CCPG |
| Yunnan | 0.4150 | SCPG | Shaanxi | 0.8696 | NWPG |
| Qinghai | 0.2263 | NWPG | Ningxia | 0.8184 | NWPG |
| Hebei | 0.9148 | NCPG | Hunan | 0.5523 | CCPG |
| Liaoning | 0.8357 | NCPG | Hainan | 0.6463 | SCPG |
| Shanghai | 0.7934 | ECPG | Guizhou | 0.6556 | SCPG |
| Anhui | 0.7913 | ECPG | Gansu | 0.6124 | NWPG |
| Shandong | 0.9236 | NCPG | Xinjiang | 0.7636 | NWPG |

The NCPG, NECPG, ECPG, CCPG, SCPG, and NWPG represent northern China power grid, northeast China power grid, eastern China power grid, central China Power grid, southern China power grid, and northwest China power grid, respectively.

**Table A3.** The detailed degree centrality and betweenness centrality of the spatial association network for China's direct RCEs in 2000 and 2019.

| Province | Degree Centrality (%) | | Betweenness Centrality | |
|---|---|---|---|---|
| | 2000 | 2019 | 2000 | 2019 |
| Beijing | 93.10 | 79.31 | 25.22 | 12.42 |
| Tianjin | 58.62 | 72.41 | 8.67 | 10.05 |
| Hebei | 6.90 | 13.79 | 0.00 | 0.03 |
| Shanxi | 13.79 | 20.69 | 0.06 | 0.11 |

**Table A3.** *Cont.*

| Province | Degree Centrality (%) | | Betweenness Centrality | |
|---|---|---|---|---|
| | **2000** | **2019** | **2000** | **2019** |
| Inner Mongolia | 10.34 | 13.79 | 0.04 | 0.06 |
| Liaoning | 10.34 | 13.79 | 0.04 | 0.06 |
| Jilin | 13.79 | 17.24 | 0.06 | 0.12 |
| Heilongjiang | 17.24 | 17.24 | 0.14 | 0.12 |
| Shanghai | 89.66 | 89.66 | 20.04 | 14.98 |
| Jiangsu | 68.97 | 82.76 | 8.38 | 12.56 |
| Zhejiang | 58.62 | 62.07 | 5.82 | 5.00 |
| Anhui | 20.69 | 24.14 | 0.16 | 0.27 |
| Fujian | 27.59 | 44.83 | 0.45 | 2.09 |
| Jiangxi | 20.69 | 24.14 | 0.16 | 0.27 |
| Shandong | 17.24 | 20.69 | 0.07 | 0.26 |
| Henan | 24.14 | 27.59 | 0.42 | 0.39 |
| Hubei | 17.24 | 48.28 | 0.09 | 1.71 |
| Hunan | 20.69 | 34.48 | 0.16 | 0.51 |
| Guangdong | 51.72 | 44.83 | 3.55 | 2.09 |
| Guangxi | 24.14 | 37.93 | 0.22 | 0.63 |
| Hainan | 20.69 | 27.59 | 0.09 | 0.34 |
| Chongqing | 24.14 | 37.93 | 0.33 | 0.60 |
| Sichuan | 20.69 | 31.03 | 0.21 | 0.51 |
| Guizhou | 24.14 | 31.03 | 0.22 | 0.51 |
| Yunnan | 20.69 | 31.03 | 0.21 | 0.51 |
| Shaanxi | 17.24 | 27.59 | 0.14 | 0.34 |
| Gansu | 31.03 | 41.38 | 1.09 | 1.31 |
| Qinghai | 20.69 | 31.03 | 0.21 | 0.44 |
| Ningxia | 17.24 | 24.14 | 0.06 | 0.12 |
| Xinjiang | 13.79 | 24.10 | 0.04 | 0.34 |
| Mean | 29.20 | 36.55 | 2.55 | 2.29 |

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
