# Peer review of "Spatiotemporal Dynamics of Direct Carbon Emission and Policy Implication of Energy Transition for China’s Residential Consumption Sector by the Methods of Social Network Analysis and Geographically Weighted Regression"

_land, doi:10.3390/land11071039_

Round 1

Reviewer 1 Report

In this paper, the authors use Social Network Analysis and Geographically Weighted Regression techniques to investigate the energy consumption situation in China. 

This paper deals with a very important issue in the current scientific and geopolitical scenario, and the approach proposed by the authors to solve it is original in that it is based on Social Network Analysis, which is generally used in contexts very different from this. The approach proposed by the authors appears to be correct and the results obtained are satisfactory.

In my opinion, the authors, with this paper, open a line of research that can also be considered in the future. In fact, they have used the basics of Social Network Analysis in their paper. In the future, they may think of adapting various advanced notions on Social Network Analysis, which have already been proposed in other fields, to their context. I suggest, therefore, that the authors add a new section, called "Discussion," immediately before the conclusion, in which they highlight this possibility. In this section, among others, they could mention the following advanced concepts of Social Network Analysis to be applied in the future to their context:

- Multi-dimensional social network-based model (introduced in "Investigating negative reviews and detecting negative influencers in Yelp through a multi-dimensional social network based model").

- Scope (introduced in "An approach to compute the scope of a social object in a Multi-IoT Scenario).

- Trust and reputation (introduced in "An approach to evaluate trust and reputation of things in a Multi-IoTs scenario")

- Colored Social Networks (introduced in "Leveraging Network Analysis to support experts in their analyses of subjects with MCI and AD")

The inclusion of such a section would make the paper even more interesting as it would open up new research perspectives for the future.

Author Response

Dear reviewer:

Thank you very much for giving us the following comments.

We deeply appreciate you for these comments because they are valuable and helpful for revising and improving our papers, as well as for our research. We have studied them carefully and have tried our best to make revisions. We marked up the revisions to the manuscript using the “Track Changes” function, such that any changes are easily viewed. The main corrections are as the follows:

Responds to the reviewer’s comments:

Comment 1: This paper deals with a very important issue in the current scientific and geopolitical scenario, and the approach proposed by the authors to solve it is original in that it is based on Social Network Analysis, which is generally used in contexts very different from this. The approach proposed by the authors appears to be correct and the results obtained are satisfactory.

Response 1: Thank you very much for this comment. We really appreciate your recognition of our study. In our study, we addressed an important issue, which is the spatiotemporal dynamics of China’s direct RCEs and the driving factors. Although Social Network Analysis (SNA) and Geographically Weighted Regression model (GWR) are widely used, no one applies two of them to deal with issues. And we are first to combine SNA with GWR to analyze the spatial associations of China’s direct RCEs and the driving factors based on comprehensive interregional association and heterogeneity of direct RCEs among provinces in China. SNA and GWR are mature methods with a comprehensive theoretical foundation. They are used by researchers in various fields, such as the field of energy consumption and carbon emissions. These provide some scientific references for our study. In addition, the results of our study can provide suggestions for the energy transition and can be compared with the findings of other researchers.

Comment 2: In fact, they have used the basics of Social Network Analysis in their paper. In the future, they may think of adapting various advanced notions on Social Network Analysis, which have already been proposed in other fields, to their context. I suggest, therefore, that the authors add a new section, called "Discussion," immediately before the conclusion, in which they highlight this possibility.

Response 2: Thank you very much for this comment. We are grateful for your valuable suggestions. In our study, we only used the basis of SNA to address some issues, which are network density, network connectedness, network efficiency, network hierarchy, degree centrality, betweenness centrality, and block model, but did not mention some advanced concepts of SNA that have been proposed in other fields. These advanced concepts can provide more research perspectives and ideas for research in our field. Thus, we highly agree with your comments and suggestions that we should discuss and highlight the possibility of adopting some advanced concepts of SNA in the section Discussion. Please see the Lines 648-650.

Comment 3: In this section, among others, they could mention the following advanced concepts of Social Network Analysis to be applied in the future to their context: Multi-dimensional social network-based model (introduced in "Investigating negative reviews and detecting negative influencers in Yelp through a multi-dimensional social network based model"); Scope (introduced in "An approach to compute the scope of a social object in a Multi-IoT Scenario); Trust and reputation (introduced in "An approach to evaluate trust and reputation of things in a Multi-IoTs scenario"); Colored Social Networks (introduced in "Leveraging Network Analysis to support experts in their analyses of subjects with MCI and AD"). The inclusion of such a section would make the paper even more interesting as it would open up new research perspectives for the future.

Response 3: Thank you very much for this comment. We are grateful for your valuable suggestions.

We studied fully these advanced concepts you mentioned and added some examples of these advanced concepts that may be used in our future study. Specifically, a multi-dimensional social network-based model can be applied to study the dissemination of the energy transition concept to public from multiple dimensions; SNA can be used to assess the scope of implementation subject of energy transition policy and the scope of policy dissemination in a Multi-IoT Scenario; SNA can also be used to evaluate trust and reputation of enterprise in achieving the goals of emissions reduction and energy transition in a Multi-IoT Scenario; the Colored Social Network can provide a fresh perspective for the research of emissions reduction and energy transition. The related revisions can be seen in Section 3.5, Lines 650-659.

In the end, we want to express our special thanks to you for your valuable comments and suggestions.

Thank you and best regards,

Yours sincerely,

Yuling Sun, Junsong Jia, Min Ju, and Chundi Chen

Corresponding author: Junsong Jia

Reviewer 2 Report

I like the paper. It is a comprehensive paper dealing with many interconnected issues and does so in a seamless manner. I have some comments, in order of appearance rather than importance:

1.       Abstract: “As China‘s second largest energy-use sector, residential consumption has a huge potential for carbon dioxide (CO2) reduction and energy saving or transition”. Give me figures. I guess the difference between the first and the second is huge.

2.       In the same vein, you wrote, “The residential sector, the second largest energy consumption sector in China, is the main source of CO2 emissions”. Why?

3.       “direct and indirect emission”. For the sake of understanding, you should elaborate on it.

4.       The justification of the sample period is naïve.

5.       Please refer to the appendix before referring to Tables A.1 and so on.

6.       Any justification for the potential driving factors?

7.       I would be more specific when it comes to explaining section 2.3

8.       The same for 2.4.1. For the authors, and expert readers, it can appear to be obvious, but, for instance, nothing is said about the concept of “nodes”, which is a general one, for this specific case study. Sometimes you’d better explain these things in layman’s terms or, otherwise, the reader can get lost.

9.       Another example: “the number of other nodes directly connected to this node”. So far, not have any clue about the way of getting this number. “the number of shortcuts that exist between nodes j and k “, the same.

10.   “Here, we applied a convergent correlations (CONCOR) method and chose an iteration criterion of 0.2 and a maximum partition density of 2 to perform block model analysis.” Please, inform the reader about these important? decisions.

11.   What is the way of setting the number of provinces in the network?

12.   I have used GWR for other issues, and the specification of equation (11) is a bit rare. Once again, I would elaborate on it. For instance, beta0(ui,vi)

13.   Y is the dependent variable, x are the independent variables. Why are you using a general specification? At least, I would mention in brackets the variables you are going to use later.

14.   Before presenting the results, I would inform the reader of the program/routines used in the calculations.

15.   It is quite hard to get any conclusions from Figure 2

16.   Page 10. Line 365. Figure 3 rather than Figure 2?

17.   Section 3.3. The analysis for 2000 would be also realistic. You simply focus your comments on the most current data.

18.   Potential endogeneity in the equation? That’s obvious.

19.   Page 14. Line 472. Figure 6.

Author Response

Dear reviewer:

Thank you very much for giving us the following comments.

We deeply appreciate you for these comments because all of them are valuable and helpful for revising and improving our papers, as well as for our research. We have studied them carefully and have tried our best to make revisions. Revised portions are marked in yellow on the revised manuscript. The main corrections are as the follows:

Responds to the reviewer’s comments:

Comment 1: Abstract: “As China’s second largest energy-use sector, residential consumption has a huge potential for carbon dioxide (CO2) reduction and energy saving or transition”. Give me figures. I guess the difference between the first and the second is huge.

Response 1: Thank you very much for this comment. Figure 1 shows the total energy consumption by sector of China from 2000 to 2019. From 2000 to 2019, the industry sector has been the largest energy-use sector; the residential sector has been the second largest energy-use sector. In 2019, the amount of energy used for industry sector was 3.225 × 109 tce; the amount of energy used for residential sector was 0.617 × 109 tce; the difference between the largest and the second largest is huge.  The Figure 1 can be seen in the attachment.

Comment 2: In the same vein, you wrote, “The residential sector, the second largest energy consumption sector in China, is the main source of CO2 emissions”. Why?

Response 2: Thank you very much for this comment. We added a figure named “Figure 1”. It shows the total energy consumption by sector of China from 2000 to 2019. It can be seen from Figure 1 that the residential sector has been the second large energy use sector from 2000 to 2019. In addition, we have mentioned “Figure 1” in a bracket right behind “The residential sector, the second largest energy consumption sector in China”. The related revisions can be seen in Line 52 and Line 85, Figure 1.    

Comment 3: direct and indirect emission”. For the sake of understanding, you should elaborate on it.

Response 3: Thank you very much for your comment and valuable suggestions. The RCEs include two concepts, which are direct and indirect emissions. Direct RCEs are CO2 emissions that directly come from residents consuming amounts of fossil energy and secondary energy in activities such as lighting, cooking, and travel by personal transport. Indirect RCEs are induced by the energy use of non-energy products consumed by residents in clothing, food, housing, and transportation for all life-cycle links. As your suggestion, we moved the definition of direct emission from Lines 194-196 to Lines 90-92. And we also elaborated on indirect emission. The related revisions can be seen in the Lines 194-196, 90-94.

Comment 4: The justification of the sample period is naive.

Response 4: Thank you very much for this comment. We are sorry that we did not give any justification of the sample period, which might make the reader confused. The main reason we set 200-2019 as the study period is the availability of data. A justification of the sample period and a justification of the key time nodes are added by us. Please see the Lines 165-168.

Comment 5: Please refer to the appendix before referring to Tables A1 and so on.

Response 5: Thank you very much. We are grateful for your valuable suggestions. As your suggestion, we mentioned the appendix before referring to Table A1 and so on. The related revisions can be seen in the Lines 174, 176, 179, 415, and 433.

Comment 6: Any justification for the potential driving factors?

Response 6: Thank you very much for this comment. Energy-related CO2 emissions are affected by economy, population, and energy consumption. This is the main reason that we chose GDP, per capita consumption expenditure (PCE), energy consumption structure (ECS), per capita energy consumption expenditure (PEC), population aging (AG), and educational level (EDU) as the driving factors of direct RCEs. Here, GDP and PCE were selected to measure economic factor; AG and EDU were chosen as population factors; ECS and PEC were selected as energy consumption factors. We moved the justification from lines 324-325 to the Data Description section. The related revisions can be seen in the Lines 183-190.

Comment 7: I would be more specific when it comes to explaining section 2.3.

Response 7: Thank you very much for this comment. We are grateful for your valuable suggestions. As you suggested, we made the following revisions to make section 2.3 more specific.

Firstly, we added the intention of direct RCEs’ spatial association network and described the meaning of node and line in the network. Please see the Lines 205-208.

Secondly, we introduced the current two main methods for portraying spatial associations and compared their advantages and disadvantages. Please see the Lines 208-213.

In addition, we described the reason that we used the share of direct RCEs of the province  in the sum of direct RCEs of the provinces  and  to correct the empirical constant, and described the meaning of . Please see the Lines 215-218, 225-226.

Comment 8: The same for 2.4.1. For the authors, and expert readers, it can appear to be obvious, but, for instance, nothing is said about the concept of “nodes”, which is a general one, for this specific case study. Sometimes you’d better explain these things in layman’s terms or, otherwise, the reader can get lost.

Response 8: Thank you very much for this comment. We appreciate your valuable suggestions. And we are sorry that we did not say about the concept of “nodes”, which makes readers get lost. The node represents the actor in network. And in our study, the node represents the province of China; the line represents the spatial association of direct RCEs between provinces. We added the meaning of node and line in the network in a different way. It can be seen in the Section 2.3, Lines 206-208.

Comment 9: Another example: “the number of other nodes directly connected to this node”. So far, not have any clue about the way of getting this number. “the number of shortcuts that exist between nodes j and k “, the same.

Response 9: Thank you very much for this comment.

To our knowledge, “The degree centrality of a node in the network indicates the number of other nodes directly connected to this node” (Line 275-276), Which is a concept of absolute degree centrality. Centrality divides into two concepts, which are absolute centrality and relative centrality. The relative centrality is a standardized form of absolute centrality. Relative centrality should be chosen when it is necessary to express the centrality of different graphs. In this study, we used relative centrality. Thus, we deleted the expression of degree centrality in our study for avoiding misleading the reader.

In addition, degree centrality and betweenness centrality can be calculated directly by software UCINET 6. These “number” can be calculated directly by UCINET 6 too. We added this clue in the Section 2.4.2, Lines 271-272. We also added the clue about the way of the number of nodes directly associated with the node, please see the Line 282.

Comment 10: Here, we applied a convergent correlations (CONCOR) method and chose an iteration criterion of 0.2 and a maximum partition density of 2 to perform block model analysis.” Please, inform the reader about these important? decisions.

Response 10: Thank you very much for this comment. We are grateful for your helpful suggestions.

A convergent correlations (CONCOR) method allows for partitioning of each actor (province) to simplify data. And the size of dataset is small in term of the number of nodes in our study. If divided above 4 clusters, there were more clusters consisting of only a few nodes. Consequently, the defective results of spatial Clustering analysis will occur. Therefore, we applied a CONCOR method to perform block model analysis, and chose an iteration criterion of 0.2 and a maximum partition density of 2 for getting 4 clusters.

We added the above contents in Lines 302-306 for informing the reader about the reason and importance of which applied a convergent correlations (CONCOR) method and chose an iteration criterion of 0.2 and a maximum partition density of 2 to perform block model analysis.

Comment 11: What is the way of setting the number of provinces in the network?

Response 11: Thanks a lot for this moment.

A spatial association network of direct RCEs is an aggregate of exploring the direct RCEs relationships between provinces. Each province is a node in the network. The number of provinces is the number of research subjects. Considering the data unavailability of Tibet’s RCEs and the inconsistent statistical caliber of Taiwan, Hong Kong, and Macao, 30 provinces that include autonomous regions and municipalities in China are used as the research subjects in this paper; Tibet, Taiwan, Hong Kong, and Macao are not included. We added these contents in Section 2.1, Lines 160-163.

Comment 12: I have used GWR for other issues, and the specification of equation (11) is a bit rare. Once again, I would elaborate on it. For instance, beta0(ui,vi)

Response 12: Thank you very much for this comment. We presented and checked the form of GWR carefully with reference to Fighteringham et al. [1] and Shen et al. [2].

GWR extends the traditional linear regression framework by allowing local rather than global parameters. It embeds spatial location of data into the regression parameters and uses locally weighted least squares methods for point-by-point parameter estimation.  is the longitude and latitude coordinates of ;  denotes the intercept parameters for location point , is a continuous function of geographical location, which denotes the k-th coefficient of the independent variable at location point . We elaborated on them in Lines 335-336; 338-440. Here is the reference.

[1] Fotheringham, A.S.; Brunsdon. C. Local forms of spatial analysis. Geogr. Anal. 1999, 31, 340-358.

[2] Shen, T.Y.; Yu, H.C. Spatial econometrics; Peking University Press: Beijing, China, 2019

Comment 13: Y is the dependent variable; x are the independent variables. Why are you using a general specification? At least, I would mention in brackets the variables you are going to use later.

Response 13: Thank you very much for this comment. We are grateful for your helpful suggestions. We used a general specification of x and Y, because we did not consider the difficulties of understanding that may result from using the general specification. We are sorry about this. As you suggested, we added the variables we are about to using later in brackets. It can be seen in Lines 335, and 337.

Comment 14: Before presenting the results, I would inform the reader of the program/routines used in the calculations.

Response 14: Thank you very much for this comment. We really appreciate for your valuable suggestions. As you suggested, we mentioned and added the routines (Equation) used in the calculations before showing results. Please see the Lines 346, 377, 389-390, 414, and 564.

Comment 15: It is quite hard to get any conclusions from Figure 2

Response 15: Thank you very much for this comment. We are sorry for our mistake that uploading two same figures, which might make readers be confused. Both the original Figure 2 (a) and Figure 2 (b) were the spatial association network of China’s direct RCEs in 2000. We uploaded the new Figure 3 (b) in the Lines 381-383. It can be seen from new Figure 3 (b) that the number of lines increased obviously compared with Figure 3 (a). This indicates that the number of associations of direct RCEs has increased over the two decades.

Comment 16: Page 10. Line 365. Figure 3 rather than Figure 2?

Response 16: Thank you very much for this comment. It was original Figure 2. Here is the reason that we put a clue for original Figure 2 instead of original Figure 3 in original line 365. We wanted to express that the decline trend of network efficiency over the study period further confirms the conclusion getting from original Figure 2. For reducing readers’ confusion, we deleted it in the revised manuscript. It can be seen in the Line 408.

Comment 17: Section 3.3. The analysis for 2000 would be also realistic. You simply focus your comments on the most current data.

Response 17: Thank you very much for this comment and suggestion. As you suggested, we added the spatial clustering analysis for 2000 and changed some corresponding contents. It can be seen in Section 3.3.

In this section, we told the number of provinces that each cluster contained (Lines 439-440), and described the type of Clusters A, B, C, and D (Lines 446-461) in 2000. We also made a summary of the spatial clustering for 2000 and 2019 (Lines 484-511). Moreover, we added a new Figure 6 (a), which is the clusters’ spatial distribution of the spatial association network for China’s direct RCEs in 2000 (Lines 511-515). Next, we compared the clusters’ spillover effect of spatial association network for China’s direct RCEs in 2000 and 2019 (Lines 519-522). In addition, we examined the transmission mechanism of the direct RCEs flow among provinces in 2000. It should be noted that we deleted one of the reasons that Cluster C played an intermediary role in the network (Lines 543-545). And we explained it carefully in Lines 496-502.       

Comment 18: Potential endogeneity in the equation? That’s obvious.

Response 18: Thank you very much for this comment. we appreciate and agree with your comment.

Endogeneity is a problem that infects an unknown portion of empirical studies. It leads to biased model estimation results [1]. GWR extends the traditional linear regression framework by allowing local rather than global parameters [2]. It also faces potential endogeneity issues [3]. Therefore, it is important to discuss endogeneity. Endogeneity occurs when an independent variable correlates with the error term of the dependent variable in a predictive model. Wooldridge separates the causes of endogeneity into four categories, which are omitted variable, simultaneity, measurement error, and selection. The second cause of endogeneity is simultaneity. The estimate of how x affects y is biased if y also affects x, which is simultaneity, the cause of endogeneity [4]. In this case, x is the endogenous variable. An instrumental variable method is an important method to address endogeneity and is used by most researchers [5]. An instrumental variable needs to be selected for the study, which should be correlated with the endogenous variable but not with other variables, and the instrumental variable can only influence the dependent variables through the endogenous variable [6].

In our study, we selected direct RCEs as the dependent variable, gross national product (GDP), energy consumption structure (ECS), population aging (AG), per capita consumption expenditure (PCE), per capita energy consumption (PEC), and educational level (EDU) as independent variables. The growth of per capita energy consumption will promote the growth of direct RCEs. And the growth of direct RCEs will promote adjustment of energy consumption structure to reduce the per capita energy consumption in the context of global warming, which is simultaneity. Obviously, PCE is the endogenous variable, and ECS is another independent variable that is closely related to PCE. Moreover, they are indicators of energy consumption. Therefore, an instrumental variable that is correlated with PEC will also correlate with ECS with a high probability. In addition, researchers using GWR in the field of energy consumption rarely discuss endogeneity [7,8-10], thus lacking a valuable scientific reference for selecting instrumental variables and discussing endogeneity.

Considering the above contents, although we did not discuss and address potential endogeneity, we added this limitation in the Discussion Section. It can be seen in Lines 679-686. Here is the reference.

[1] Arbia, G. A primer for spatial econometrics with applications in R; Palgrave Macmillan Press: London, UK, 2014

[2] Fotheringham, A.S.; Brunsdon. C. Local forms of spatial analysis. Geogr. Anal. 1999, 31, 340-358.

[3] Mainardi, S. Modelling spatial heterogeneity and anisotropy: Child anaemia, sanitation and basic infrastructure in sub-Saharan Africa. Int. J. Geogr. Inf. Sci. 2012, 26, 387-411.

[4] Wooldridge, J. M. Econometric analysis of cross section and panel data; MIT Press: Cambridge, UK, 2010.

[5] Hill, A.D.; Johnson, S.G.; Greco, L.M.; O’Boylee, E.H.; Walter, S.L. Endogeneity: A review and agenda for the methodology-practice divide affecting micro and macro research. Journal of Management. 2020, 47, 105-143.

[6] Hill, R.C.; Griffiths, W.E.; Lim, G.C. Principles of econometrics; Wiley Press: Hoboken, US, 2007.

[7] Wang, Y.N.; Zhao, M.J.; Chen, W. Spatial effect of factors affecting household CO2 emissions at the provincial level in China: a geographically weighted regression model. Carbon Manag. 2018, 9, 187-200.

[8] Chen, X.H.; Zhao, Q.; Huang, F.; Qiu, R.Z.; Lin, Y.H.; Zhang, L.Y.; Hu, X.S. Understanding spatial variation in the driving pattern of carbon dioxide emissions from taxi sector in great eastern China: Evidence from an analysis of geographically weighted regression. Clean Technol. Envir. 2020, 22, 979-991.

[9] Qin, H.T.; Huang, Q.H.; Zhang, Z.W.; Lu, Y.; Li, M.C.; Xu, L. Carbon dioxide emission driving factors analysis and policy implications of Chinese cities: Combining geographically weighted regression with two-step cluster. Sci. Total Environ. 2019, 684, 413-424.

[10] Yang, X.H.; Jia, Z.; Yang, Z.M.; Yuan, X.Y. The effects of technological factors on carbon emissions from various sectors in China-A spatial perspective. J. Clean. Prod. 2021, 301.

Comment 19: Page 14. Line 472. Figure 6.

Response 19: Thank you very much for this comment. Original Figure 6 can show the variation of regression coefficients of six driving factors across provinces in 2000 and 2019. Since the change of regression coefficients of six driving factors is small, it might be hard to see the spatiotemporal variation for each province. But the overall trend of regression coefficients is obvious. In addition, the regression coefficients of EDU are very small, and the colour is very light for illustrating these regression coefficients. It might be difficult for readers to distinguish it from the background of manuscript. Thus, we added a border for this figure. Please see the Line 587.   

In the end, we want to express our special thanks to you for your valuable comments, questions, and suggestions.

Thank you and best regards,

Yours sincerely,

Yuling Sun, Junsong Jia, Min Ju, and Chundi Chen

Corresponding author: Junsong Jia

Round 2

Reviewer 1 Report

In my previous review, I requested the Authors to add a section called "Discussion" where indicating several new horizons of their research. I also suggested some papers to consider in that section. Unfortunately, no section "Discussion" has been put and none of those paper has been considered.

The authors added 2 ROWS!!!!! instead of a section. This is not serious.

If possible, I suggest to give them a last chance (major revision) to perform a serious review with my requests. If this is not possible I suggest to reject this paper.

Author Response

Dear reviewer:

Thank you very much for giving us the following comment.

We deeply appreciate you for this comment because it is valuable and helpful for revising and improving our papers, as well as for our research. We have studied it carefully and have tried our best to make revisions. Revised portions are marked in yellow on the new manuscript. The main corrections are as the follows:

Responds to the reviewer’s comment

Comment: In my previous review, I requested the Authors to add a section called "Discussion" where indicating several new horizons of their research. I also suggested some papers to consider in that section. Unfortunately, no section "Discussion" has been put and none of those paper has been considered. The authors added 2 ROWS!!!!! instead of a section. This is not serious. If possible, I suggest to give them a last chance (major revision) to perform a serious review with my requests. If this is not possible, I suggest to reject this paper.

Response: Thank you very much for this comment. We appreciate the last chance you gave us. We are sorry that we did not understand fully your kind suggestions for our previous revision. And we are sorry that section “Discussion” hasn’t been put and none of papers you suggested have been considered. In this revision, we studied your helpful suggestions carefully and added a new section “Discussion”. In this section, we not only kept what we have discussed in original manuscript but also added some new discussions. The papers you suggested also were considered in this revision. We listed some advanced concepts of social network analysis, which are multi-dimensional social network based-model, the concept of scope in a Multi-IoT scenario, trust and reputation in a Multi-IoT scenario, colored social networks, and social network analysis-based classifier (SNAc). Multi-dimensional social network based-model, the concept of scope in a Multi-IoT scenario, trust and reputation in a Multi-IoT scenario, and colored social networks are the concepts introduced in the papers you suggested. We added the aspects of these concepts that may be used in our future work. The related revisions can be seen in section Discussion, Lines 638-673.

In the end, we want to express our special thanks to you for your valuable comments and suggestions.

Thank you and best regards,

Yours sincerely,

Yuling Sun, Junsong Jia, Min Ju, and Chundi Chen

Corresponding author: Junsong Jia

Reviewer 2 Report

The paper has certainly improved. Congratulations.

Only some minor comments related to previous ones:

1.       Related to my previous comment, you have to admit that the difference between the first and the second is huge.

2.       Related to my previous comment on the sentence, “The residential sector, the second largest energy consumption sector in China, is the main source of CO2 emissions”. I insist, why is it the main source of CO2? Being the difference with the industrial sector so big, it is something that the reader is bound to wonder.

3.       I am sorry to insist on the justification for the potential driving factors. What you did has nothing to do with a justification.

Author Response

Dear reviewer:

Thank you very much for giving us the following comments.

We deeply appreciate you for these comments because all of them are valuable and helpful for revising and improving our papers, as well as for our research. We have studied them carefully and have tried our best to make revisions. Revised portions are marked in yellow on the revised manuscript. The main corrections are as the follows:

Responds to the reviewer’s comments:

Comment 1: Related to my previous comment, you have to admit that the difference between the first and the second is huge.

Response 1: Thank you very much for this comment. From 2000 to 2019, the difference in total energy consumption between residential sector and industrial sector is huge. The industry sector has been the largest energy-use sector; the residential sector has been the second largest energy-use sector. We admitted the difference in the Introduction. The related revisions can be seen in Lines 54-55.  

Comment 2: Related to my previous comment on the sentence, “The residential sector, the second largest energy consumption sector in China, is the main source of CO2 emissions”. I insist, why is it the main source of CO2? Being the difference with the industrial sector so big, it is something that the reader is bound to wonder.

Response 2: Thank you very much for this comment. We are sorry that we used an inappropriate word to express our ideas. We used the word “non-negligible” instead of the word “main” in this revision. From 2000 to 2019, the industry sector has been the largest energy consumption sector; the residential sector has been the second largest energy consumption sector. Energy consumption is closely related to CO2 emissions. Therefore, the residential sector is the non-negligible source of CO2 emissions. The related revisions can be seen in Lines 53-57.   

Comment 3: I am sorry to insist on the justification for the potential driving factors. What you did has nothing to do with a justification.

Response 3: Thank you very much for your comment. We are sorry that we did not give enough justifications for the potential driving factors. We studied your suggestions carefully and added the justifications for the potential driving factors. The related revisions can be seen in lines 186-213.

In the end, we want to express our special thanks to you for your valuable comments, questions, and suggestions.

Thank you and best regards,

Yours sincerely,

Yuling Sun, Junsong Jia, Min Ju, and Chundi Chen

Corresponding author: Junsong Jia

Round 3

Reviewer 1 Report

This time the authors have complied with my requests. Therefore, in my opinion, the paper can be accepted.

Reviewer 2 Report

Thanks for your effort